# A trustworthy hybrid model for transparent software defect prediction: SPAM-XAI

**Mohd Mustaqeem**[1], **Suhel Mustajab**[1], **Mahfooz Alam**[1]\*, **Fathe Jeribi**[2]\*, **Shadab Alam**[2], **Mohammed Shuaib**[2]

1 Department of Computer Science, Aligarh Muslim University, Aligarh, India, 2 Department of Computer Science, College of Engineering and Computer Science, Jazan University, Jazan, Saudi Arabia

\* fjeribi@jazanu.edu.sa (FJ); mahfoozalam.amu@gmail.com (MA)

**Data Availability Statement:** All relevant data are within the manuscript.

**Funding:** The authors extend their appreciation to the Deputyship for Research& Innovation, Ministry

## Abstract

Maintaining quality in software development projects is becoming very difficult because the complexity of modules in the software is growing exponentially. Software defects are the primary concern, and software defect prediction (SDP) plays a crucial role in detecting faulty modules early and planning effective testing to reduce maintenance costs. However, SDP faces challenges like imbalanced data, high-dimensional features, model overfitting, and outliers. Moreover, traditional SDP models lack transparency and interpretability, which impacts stakeholder confidence in the Software Development Life Cycle (SDLC). We propose SPAM-XAI, a hybrid model integrating novel sampling, feature selection, and eXplainable-AI (XAI) algorithms to address these challenges. The SPAM-XAI model reduces features, optimizes the model, and reduces time and space complexity, enhancing its robustness. The SPAM-XAI model exhibited improved performance after experimenting with the NASA PROMISE repository's datasets. It achieved an accuracy of 98.13% on CM1, 96.00% on PC1, and 98.65% on PC2, surpassing previous state-of-the-art and baseline models with other evaluation matrices enhancement compared to existing methods. The SPAM-XAI model increases transparency and facilitates understanding of the interaction between features and error status, enabling coherent and comprehensible predictions. This enhancement optimizes the decision-making process and enhances the model's trustworthiness in the SDLC.

## 1. Introduction

Software quality assurance plays a critical role in successfully developing and deploying software applications. Early detection of software defects is essential for mitigating risks, reducing costs, and ensuring overall project success. However, traditional SDP models face various challenges, including imbalanced datasets, high-dimensional feature spaces, and a lack of transparency in decision-making. These limitations hinder the effectiveness and reliability of SDP models in real-world applications. The high dimensionality with hundreds of variables creates an overfitting problem. Besides, the model's non-transparent decision system worsens trust and real-life application prospects. The presence of outliers and the complexity of the feature

of Education in Saudi Arabia, for funding this research work through the project number ISP-2024. The funders had no role in study design, data collection and analysis, decision to publish, or preparation of the manuscript."

**Competing interests:** The authors have declared that no competing interests exist.

space intensify the challenges that standard models face in effectively predicting defects. Therefore, this research aims to address these challenges by proposing a novel SDP model, termed SPAM-XAI, which leverages advanced machine learning (ML) techniques to enhance prediction accuracy and transparency.

The existing SDP models for software engineering are mainly based on a hands-on approach, which is not always effective. The data utilized by this paradigm is frequently inherently biased or skewed in the case of sample distribution across its accessible classes. These are datasets with many times more perfect instances than imperfect ones, if any. When these datasets are contributed to the training datasets for predictors, there is the risk of experiencing over-fit situations for the models [1]. The authors of [2] have used the Grey Wolf Optimization and Multilayer Perception (GWOFS-MLP) for SDP. In [3], authors have balanced datasets to get good results. The [4] used optimization, feature selection, dropout, and autoencoder concepts, but still, there are possibilities of non-guarantee on parameter settings and convergence. Plus, the whole thing is costlier to train due to using a complex data model that can add up the cost. Based on the viewpoint of [5], the Microtext-Deep Neural Networks technique is applied in the dynamic fault prediction of mobile apps with JIT; however, since MTL-DNN can be complex and may require you to have a well-tailored hyperparameter tuning to achieve a good outcome. In [6], the author proposes a layered recurrent neural network (L-RNN) architecture for SDP that employs an iterative extraction feature model, resulting in an excellent ROC-AU curve. However, it is a complex and slow training [7]. The author applies the node2defect method to enhance defect prediction accuracy by 9.15% using network embeddings and traditional features, but it has a high computational cost for network embedding [8]. According to [9], overlaps in the class impact on SDP on applying the K-means data cleaning technique emerged. Still, K-means clustering affects through outliers. In [10], the author developed a mathematical, cost-effective model of an SDP model. The author highlighted the consequences of buffer overflows by focusing on 15 ML algorithms and 4000 defects in the work [11]. When dealing with noisy data and outliers, the Adaboost algorithm becomes ineffective and inaccurate in SDP conditions. Although neural networks can mitigate outliers using activation functions, they remain error-prone [12].

The MLP is demonstrated as a highly effective technique, which showed superior performance compared to other approaches [13]. The proposed model also has high flexibility compared to the Gaussian Naïve Bayes classifier, which is mainly suitable for large-scale datasets and outperforms small datasets. Moreover, the limitation of the k-nearest neighbors (K-NN) algorithm is that it keeps all the training data and is, therefore, slow and prone to being disturbed by the outliers. Considering these traits of classifiers, researchers can make decisions to apply specific challenges of the SDP process being analysed.

We have proposed the SPAM-XAI model to overcome the above-mentioned limitations and problems. This model provides a robust and compact approach to improving the SDP. The SPAM-XAI model solves the problems of imbalanced datasets and the high feature space dimensionality, which the classical SDP models mainly deal with. The SPAM-XAI model applies carefully fine-tuned hyperparameters to ensure optimal performance of such a model with explanation and transparency by which a decision-making process is operated, and then with the information on factors that contribute to its prediction becomes much clearer. In comparison, the conventional SDP model may struggle with these challenges. This eventually results in improved SDP performance in the SDP. This approach gives software development brigades usable results which help reveal defects at the early stages of SDLC and make bridging the gap between powerful prediction models and human understanding possible. The SPAM-XAI model is primarily integrated into the testing phase of the SDLC. However, its influence extends to other stages, including design, implementation, and maintenance, enhancing the overall software development process. Moreover, the SPAM-XAI model is an exceptional and

productive instrument for assuring flawless, defect-free software projects and uplifting the standard of software development practices.

- The SPAM-XAI model is a hybrid approach in SDP that solves the challenges, such as dealing with imbalanced data, high-dimensional features, model overfitting, outliers, and lack of transparency using various XAI techniques for the development of a deep understanding of its decision-making processes and establishing trust among stakeholders in the SDLC.

- The SPAM-XAI model integrates the Synthetic Minority oversampling technique (SMOTE) for oversampling, Principal Component Analysis (PCA) for feature selection, and MLP as the classifier. Unlike many models that rely on linear functions or discard noisy data, SPAM-XAI employs a logistic function as an activation function, directly addressing this challenge. This approach yields promising results across various datasets from NASA's directory, including CM1, PC1, and PC2.

- The SPAM-XAI model has computationally efficient and optimal training. Unlike those models in the literature, by increasing expenses because they train on the high dimensional complex datasets using intricate models, our model can reduce time complexity and optimize feature values.

- Our SPAM-XAI model surpasses the performance of other baseline models, Naïve Bayes (NB), SVM, Random Forest (RF), Logistic Regression (LR), Decision Tree (DT), K-Nearest Neighbors (KNN), Linear discriminant analysis (LDA), and Quadratic Discriminant Analysis (QDA) in both datasets - CM1 and PC1. In addition, SPAM-XAI reaches remarkable performance for the CM1; the accuracy is 98.20%, and the result is 97.51% for PC1, showing its strength and effectiveness.

- The SPAM-XAI model showcased superior performance across CM1 and PC1 datasets, exhibiting higher precision, recall, f-measure, accuracy, and au-roc than previous models. Our proposed method outperformed various algorithms, including SVM, RF, Immunos, HSOM, Artificial Neural Network-artificial bee colony (ANN-ABC), NB, C4.5 Miner, Majority Vote, AntMiner+, ADBBO-RBFNN, DT, KNN, and MLP in various areas of the experiment. The superiority of SPAM-XAI to the traditional approaches by which the systems are built is demonstrated, making this technology a robust and advanced solution.

The remaining part of this paper is structured as follows. Section 2: "Literature Review" – This section of the paper studies the literature surrounding concepts. Section 3, "Comparative Analysis of Various Algorithms with the SPAM-XAI Model", compares the SPAM-XAI model and other algorithms. Section 4, "Proposed Modelling", describes the proposed SPAM-XAI model and its components or methods used in modelling. Section 5, "Implementation", demonstrates the implementation process of the SPAM-XAI model. Section 6: "Results and Discussion Section", this section includes the description of the experimental results. Section 7: "eXplainability", in this section the work performed to implement XAI techniques within SPAM-XAI is presented. At last, Section 8, i.e., "Conclusion and Future Direction", concludes the findings of the study and outlines the scope for further research. This organization helps to present realistic data about the SDP and use a clear and comprehensible SPAM-XAI model to explain the accuracy of its interpretation.

## 2. Literature review

As we know, with the daily growth of the software industry, more cost, time, and effort are required to test the software, which increases maintenance costs and achieves software quality

assurance. So, there is a requirement to develop the SDP models and use statistical data to predict the faulty modules so that the developers can use their skills in designing and not in testing [14,15]. The potential influence of this work on software development methodologies makes it significant. By improving the accuracy and interpretability of SDP, the SPAM-XAI model can help developers identify defective modules early, leading to enhanced software quality and reduced maintenance costs. Moreover, adopting XAI techniques contributes to the growing field of XAI, where understanding the decision logic of AI models is crucial in applications such as SDP. The empirical findings from evaluating the SPAM-XAI model on real-world datasets demonstrate its superiority over traditional SDP models, validating its effectiveness in addressing imbalanced data and providing transparent outcomes. The empirical study of previously developed SDP models & techniques can be represented in this section.

## 2.1 Software defects

Bugs are also known as software defects which are characterized as mistakes or failures in a software program that lead to poor functionality or failure of the program to perform its intended task. These defects may stem from various factors such as errors that may have been coded, errors that may have been in the design of the application or may have been in the requirement specifications. Software defects can be classified into three major types: Nature based software defects, Priority based software defects, and Severity based software defects [16] as shown in Fig 1. Software defects can be classified into three types such as nature, priority, and severity. These include different features and effects of defects within each category, yet all offer valuable insights that help in acquiring essential knowledge on managing defects.

Nature-based Software Defects: Functional Bugs: Defects causing software malfunctions. Metrics include "Defect Density" and "Requirement Coverage", Unit-level bugs: Issues related to specific software units. Measured by Unit "Test Coverage" and "Defect Leakage", Integration Level Bugs: Arise from combining multiple components. Metrics include "Integration Test Coverage" and "Defect Escape Rate", Usability Defects: Affect user experience. Evaluated through "Usability Testing Scores" and "Task Success Rate", Performance Defects: Impact efficiency like response time. Metrics include "Response Time" and "Resource Utilization", Security Defects: Relate to vulnerabilities. Measured by "Vulnerability Count" and "Mean Time to Patch", Compatibility Defects: Issues with device or software compatibility. Metrics include

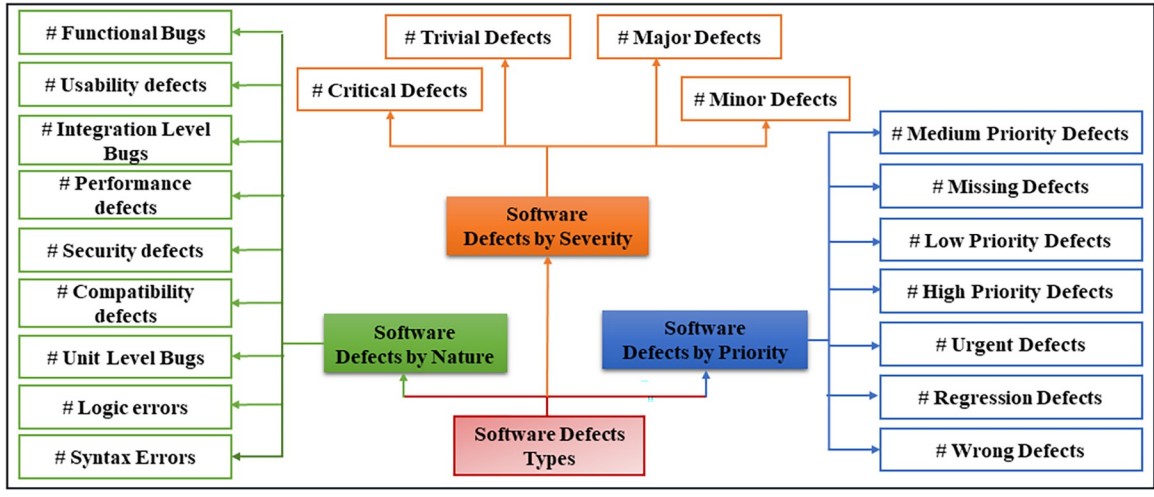

**Fig 1. Software defects types.**

"Compatibility Test Coverage" and "Compatibility Issue Ratio", Syntax Errors: Deviations from programming syntax. Assessed through "Static Code Analysis Metrics", Logic Errors: Flaws in logic. Evaluated by "Code Coverage" and "Defect Discovery Rate".

Priority-based Software Defects: Low-Priority Defects: Minor impact on operation. Metrics include "Defect Density" and "User Impact Rating", Medium Priority Defects: Can be addressed later. Measured by "Defect Resolution Time" and "Severity Rating", High-Priority Defects: Seriously affect operations. Metrics include "Defect Escalation Rate" and "Defect Resolution Efficiency", Urgent Defects: Must be fixed within 24 hours. Measured by "Urgent Defect Resolution Time" and "Escalation Rate", Missing Defects: Unmet requirements. Evaluated through "Requirement Coverage" and "Requirement Traceability Matrix", Wrong Defects: Incorrect implementation of requirements. Metrics include "Requirement Correctness" and "User Feedback", Regression Defects: Caused by code changes. Evaluated by "Regression Test Coverage" and "Defect Leakage Rate".

Severity-based Software Defects: Critical Defects: Major impact on functionality. Metrics include "Defect Impact Analysis" and "Time to Resolution", Major Defects: Significantly affect functionality. Measured by "Defect Density" and "Severity Distribution", Minor Defects: Minor functional impact. Metrics include "Defect Aging" and "User Impact Rating", Trivial Defects: No functional impact. Evaluated by "Defect Ratio" and "Defect Discovery Rate".

## 2.2 Imbalanced class problem

The imbalanced class problem occurs when the dataset is non-uniformly distributed among the classes. When ML classification algorithms are applied to these datasets, their prediction performance decreases, and models lead to overfitting. Previous research efforts have attempted to address this challenge through various balancing approaches. However, these techniques have shown limited success in significantly improving prediction accuracy [17]. According to [18], the author has used the N-US algorithm to improve SDP accuracy and AUC significantly, outperforming other models on NASA datasets, but the potential information loss from under-sampling. For instance, while multi-objective SDP models coupled with innovative optimization algorithms have been proposed to handle under-sampled datasets, scalability remains a crucial concern [19]. In [20], the authors combined ensemble learning and various ML techniques to enhance the model's performance, but they did not address the imbalance of data problem.

## 2.3 Previously developed SDP models using AI techniques

Researchers have proposed various ML-based SDP models to enhance the accuracy and efficiency of defect identification in software development. In [21], authors utilized PCA, DA, LR, and L-N Non-twenty-seven projects to evaluate performance based on predictive validity, quality, and verification cost. Similarly, in [22], the HMOCS-US-SVM model was introduced to address class imbalance and parameter selection challenges in SDP. However, limitations of this model include specific conditions for optimal performance and its efficacy across diverse datasets. In [23], a modified objective cluster analysis (OCA) was proposed for SDP, although with limited generalizability.

Furthermore, [24] introduced an optimization model using a cost-sensitive radial basis function with stacked generalization-based SDP. In a comparative study, [25] demonstrated that SVM outperformed ANN on various NASA datasets. Deep learning techniques were explored in [26], allowing for automatic learning of complex patterns without manual feature extraction. However, [27] noted the computational complexity associated with long short-term memory (LSTM) networks, posing resource challenges for large-scale projects.

Innovative approaches such as gated hierarchical LSTM networks were proposed in [28], though computational complexity remained a concern, particularly for extensive projects. Authors in [29] developed an effective model using AI-based techniques on the NASA MDP repository dataset, achieving promising results with adaptive neuron fuzzy inference system, SVM, and ANN. The FILTER technique proposed in [30] improved SVM-based SDP accuracy, although with sensitivity to hyperparameter tuning and overfitting.

Tree-based bagging ensembles were introduced in [31], but computational expenses were highlighted as a limitation. A framework eliminating the need for separate feature extraction tools was proposed in [32], although concerns about capturing crucial features arose. Convolutional neural networks (CNN) were employed [33] to enhance prediction performance, while SVM was utilized to avoid overfitting problems.

The challenge of predicting defects without labeled modules was addressed in [34] using a genetic algorithm-based LSTM-AST model. Additionally, a deep belief network (DBN)–based semantic features model was proposed in [35], which is affected by computational expenses. Despite advancements, [32] underscored the importance of reading software source code, suggesting a reliance on traditional techniques. In [36], limitations of the PCA-SVM hybridization technique and the slow convergence rate of the multi-verse optimizer algorithm (MOA) were acknowledged. By weighing these strengths and limitations, researchers can make informed decisions when selecting suitable methods for SDP and optimization tasks.

## 2.4 ML-based techniques

In [37], a NB classifier-based SDP model with normalization and noise reduction was proposed, yielding promising results. However, limitations were observed due to the limited independence assumption of the NB algorithm, which may not accurately capture interdependencies among attributes [38]. In [39], three defect prediction ML models based on the C4.5 algorithm were developed, enhancing prediction accuracy by reducing DT. Despite improvements, the model's performance was hindered by its slow execution and lack of a method for quantifying case relevance.

Biological and immunological concepts inspired by the artificial immune system were employed in SDP [40], improving defect module detection results. Although the model exhibited better recall measures, it demonstrated low evaluation metrics. In [41], a hybrid technique combining ANN-ABC algorithms was proposed, achieving successful results in defect prediction. However, the model's limitations included time-consuming training and the requirement for extensive data with multiple layers.

A semi-supervised ML-based Hybrid Self-Organizing Map (H-SOM) approach for automated SDP was introduced in [42], demonstrating good performance even without quality data. Nonetheless, challenges in obtaining accurate data and dealing with incomplete information were noted. In [43], SVM was employed as a RELIEF technique for SDP, enhancing performance metrics. However, the model's suitability for large datasets and noisy data was questioned.

An ensemble approach using weighted majority voting techniques for SDP was proposed in [44], aiming to improve performance on imbalanced datasets. However, the constraint of identical contributions from each model raised concerns about adaptability to various circumstances. In [45], a hybrid approach combining adaptive dimensional biogeography-based optimization (ADBBO) and radial basis functional neural network (RBFNN) was employed, achieving good performance metrics. However, increasing complexity with the number of neurons was observed as a limitation.

**Table 1. Comparative analysis with PCA vs feature optimization approaches.**

| S. No. | Feature Optimization Approach | Execution Speed | Performance Improvement | Feature Selection Capability |
|---|---|---|---|---|
| 1 | Genetic Algorithm | Lower complexity; Built-in feature selection | High | Yes |
| 2 | Correlation Threshold | Manual implementation; Risk of exclusion of crucial attributes | Moderate | No; Eliminates redundant features |

Parameter adjustment in K-NN significantly influenced defect prediction performance in [46], with better results obtained using Dilca distance over Euclidean distance. Nevertheless, challenges persisted in handling large datasets and feature scaling.

## 2.5 Limitations of previously developed models

Using a high-quality dataset from a large software repository can improve model performance. However, there is still a strong need for a suitable defect prediction method.

### 2.5.1 Imbalanced datasets and extreme bias.

- Most ML-based techniques that store recurrent attributes are not feasible for removing errors from the dataset [47]. Supervised algorithms are more beneficial for defect predicting at a comparable logical level. However, they are inappropriate for high-level software modules. The current SDP using classifier algorithms are somewhat incorrect in practical applications since they consider many features [36].

### 2.5.2 Accurate defect prediction model.

- Accurate, robust, noisy data for the SDP model is essential for huge projects. They utilize traditional DT for classification. However, there are several limitations to the conventional DT technique. The appropriate prediction approach is necessary to predict software defects from the archive. Data migration helps software quality approaches be more productive [48].

### 2.5.3 Consistent ML-based mechanism.

- A big challenge is the lack of a reliable ML-based technique to develop the most appropriate SDP model. Due to a lack of thorough comparison analyses of popular methodologies, testers and quality assurance specialists struggle to identify effective defect prediction models [49].

## 3. Comparative analysis of various algorithms with the SPAM-XAI model

In this section, a comparative analysis of multiple techniques is shown. Table 1 shows the feature-optimized algorithms compared with PCA. In Table 2, MLP is compared with classification algorithms, and Table 3 shows the comparison of SPAM-XAI with previously developed models.

The GA is suitable for the feature's selection and optimization but PCA was selected due to its simplicity and faster execution which is appropriate for our less complex dataset. PCA

**Table 2. MLP vs classification algorithms.**

| S. No. | Classification Method | Dataset Size | Overfitting Resistance | Visualization Ease | Computational Complexity |
|---|---|---|---|---|---|
| 1 | DT | Small | Moderate | Low | Low |
| 2 | RF | Large | High | Moderate | High |

**Table 3. SPAM-XAI compared with previous models.**

| S. No. | Algorithm | Shortcoming |
|---|---|---|
| 1 | AdaBoost | Outlier influence: Efficient handling by MLP |
| 2 | Cart | Limited adaptability; Outperformed by MLP |
| 3 | KNN | High misclassification rate compared to MLP |
| 4 | Chao Genetic | Best solution provided by PC |
| 5 | E-M model | Suboptimal solution; Guaranteed solution by SPAM-XAI |

improves performance efficiently with its embedded selection of features [50]. Setting a correlation cut-off might lead to omission of important features. PCA saves time as it removes redundancy and hence increases the performance and reliability of the feature selection [51].

DT is fast with small samples and reasonably capable of avoiding overfitting. However, they are not as useful in the aspect of offering the ease of visualization as MLPs. As far as their computational complexity is concerned, they are appropriate for simpler tasks [48]. RF work effectively with a big amount of data and have a high degree of protection against over-fitting. but they are more computationally complex and provides moderate ease of visualization. Even though CNNs are highly robust, compared to MLPs they are complex and are not easily interpreted, making them effective for specific tasks [52].

The presence of outliers also affects AdaBoost in a negative way. On the other hand, the MLP within the SPAM-XAI model is better equipped to deal with outliers thus producing more accurate predictions. The CART algorithm has low flexibility, especially with linear data. The MLP in the SPAM-XAI model is superior to CART because it can handle linear as well as non-linear datasets. The MLP is thus more accurate in classification than the KNN in the SPAM-XAI model. This makes MLP a better candidate for SDP. The Chao Genetic algorithm offers the best solutions, on the other hand, the PC approach within SPAM-XAI focuses on the best solutions by improving feature selection and optimization. The E-M model sometimes does not provide an optimal solution. However, the SPAM-XAI model provides a guaranteed solution, which enhances reliability and efficacy in SDP [53]. To address the limitations specified above, we have developed the SPAM-XAI model.

# 4. Proposed modelling

In this section, the SPAM-XAI model is proposed to enhance performance with a higher categorization rate for SDP.

## 4.1 Data description and data pre-processing

The datasets contain software measurements as attributes, along with indications of defects provided by the repository [54]. The Metrics information system is responsible for collecting and validating the data stored in the system. The authors have used the NASA MDP repository dataset to conduct an experimental investigation for SDP. The authors used CM1, PC1, and PC2 datasets that can be divided into test and train sets, and attributes are shown in Table 4.

**4.1.1 Original dataset.** The dataset taken from the NASA MDP repository is CM1, PC1, and PC2. The CM1 dataset is a NASA spacecraft instrument data written in C-programming language. PC1 and PC2 are also NASA metrics datasets, the data from earth-orbit spacecraft flight software written in C-programming language. Initially, this data was in code and converted into the numeric form using Halstead and McCabe feature extractor software metrics. These attributes were recognized in the 70s during features code characterization related to

**Table 4. PROMISE defects prediction attribute aspects.**

| Attribute name | Description of attribute |
|---|---|
| v(g) | measurement Cyclomatic complexity (McCabe) |
| uniq_opnd | unique operand overall |
| *Locomment* | *software module line comments* |
| Iv(g) | Analysis of design complexity (McCabe) |
| T | Estimator of Time |
| Total_opnd | operands total no. |
| Ev(g) | McCabe essential complexity |
| LOC | the total number of lines in the module is counted. |
| *Loblank* | *blank lines totally in the module* |
| N | The software module has a certain number of operators. |
| *uniq_op* | *unique operators overall* |
| D | difficulty Measurement |
| Branchcount | Branch total software module |
| B | Effort Estimation |
| Locodeandcomment | lines of code and comments totally |
| total_op | operators total no. |
| L | length of Program |
| E | effort with Measurement |
| I | Measurement of Intelligence |
| V | Volume |
| Defects/Problems | Information on the problem, whether the defect is present |

software quality, as shown in Table 5. Fig 2 shows the Defective ("D") and Non-Defective ("ND") in graphical form using CM1, PC1 and PC2 datasets.

The CM1 dataset contains 1988 observations. Among them, 97.6% are non-defective, and 2.4% are defective. The authors have split it into train and test sets for feeding the developed model (1391 training and 597 tests). In PC1, 97.8% of observations are non-defective, and 2.1% are defective. It contains 705 observations, which have also been divided into two parts (493 training & 212 tests) for the same. Similarly, the PC2 dataset contains 745 elements with 97.8 "ND" and 2.2% "D" It contains 745 observations, which have also been divided into two parts: 521 training & 224 tests.

## 4.2 SPAM-XAI model

The SPAM-XAI model reduces features, optimizes the model, and reduces its time and space complexity, enhancing its robustness. The working of the model can be illustrated in Fig 3, which shows the SMOTE implementation; the authors have datasets represented by the grey diagram and split data into train and test sets, as indicated by the blue and red charts. The training data is given to the SMOTE, i.e., data from the minority class set A (Rand-N)); the Euclidean distance is calculated using K-NN. The imbalanced data determines the testing rate

**Table 5. Dataset detail division.**

| Dataset | Language | Total Element | ND | D | ND % | D % |
|---|---|---|---|---|---|---|
| CM1 | C | 1988 | 1942 | 46 | 97.6 | 2.4 |
| PC1 | C | 705 | 644 | 61 | 91.3 | 8.7 |
| PC2 | C | 745 | 729 | 16 | 97.8 | 2.2 |

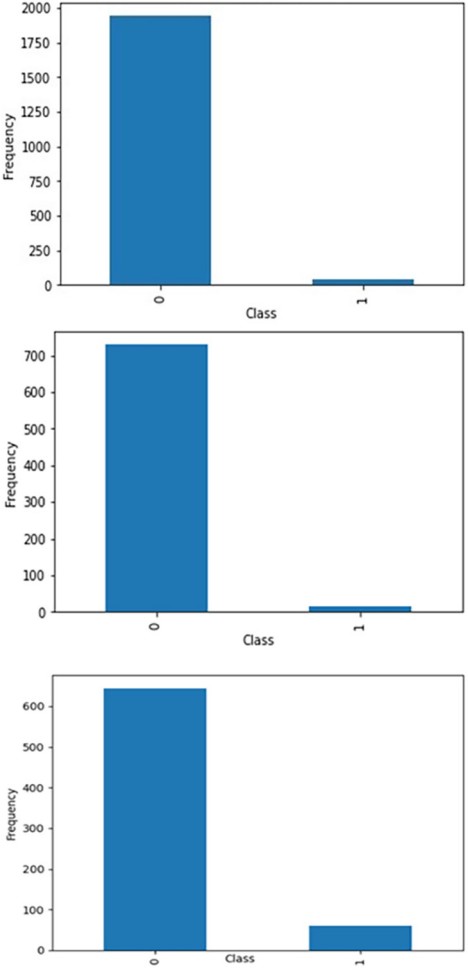

**Fig 2.** **(a).** Dataset: CM1. **(b).** Dataset: PC1. **(c).** Dataset: PC2.

N; the N models selected randomly from the closest K-NN can generate the set. The set generated by the K-NN is utilized to create a new model that will deal with the varying numbers under 0 and 1and. Finally, the desired result is sent to the next part of the proposed model.

Furthermore, this section represents the balance data generated by the SMOTE that is now fed to the PCA. Fig 3 shows that the PCA will take the training data generated by the SMOTE. This data is highly dimensional; Its dimensionality can be reduced using the mathematical covariance matrix. The Eigenvalues and Eigenvectors are also calculated. Moreover, Eigen Value Proportion (EVP) is computed for the ratio, and based on EVP, the necessary attributes are selected.

Additionally, this reduced dimensional data with the test data is provided to the MLP classifier, as shown in Fig 4. It will use multiple layers with weights; it consists of the input layer on which the data is given weights and some hidden layers. Every layer feeds to the next layer based on its computational output and continues with all the hidden layers. It uses forward propagation with an activation function (sigmoid) to optimize the values; with several iterations, the output value is compared with the original value. The following process, backpropagation, begins to adjust the weights during learning iteratively, and its main motive is to

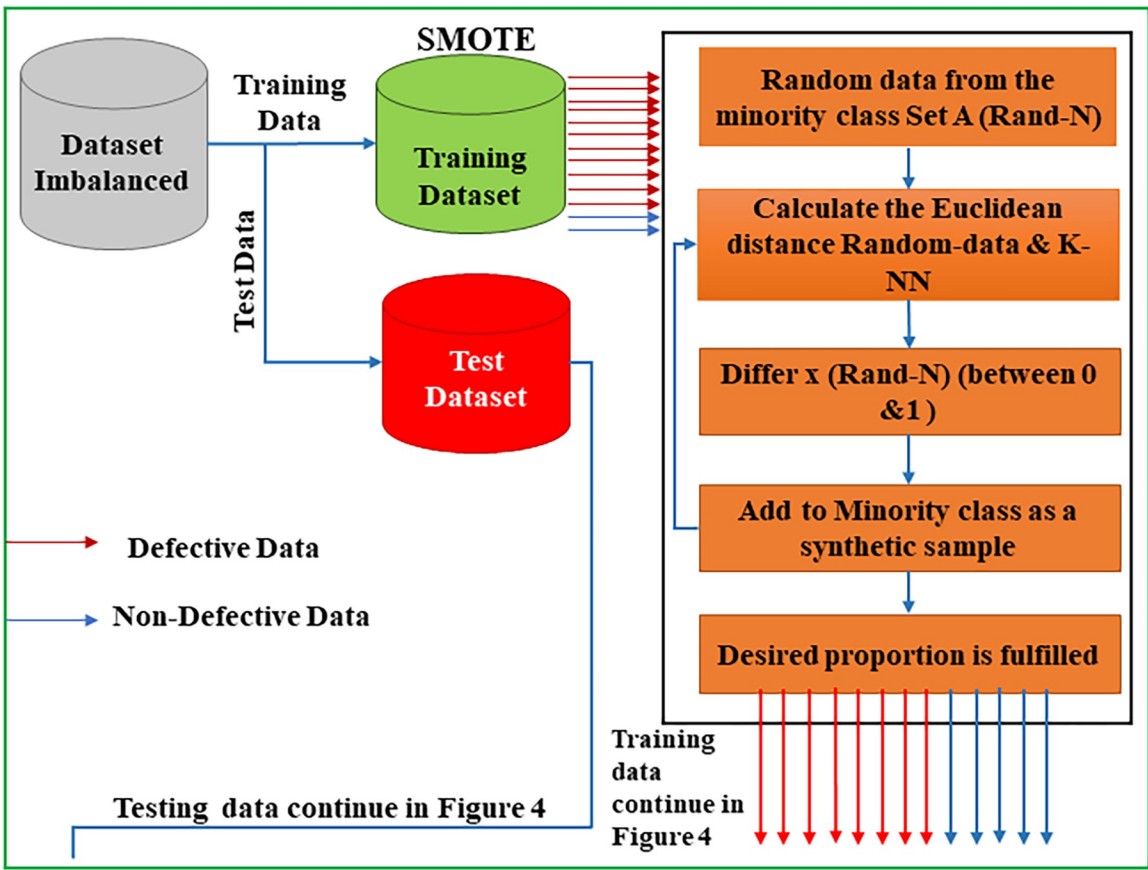

**Fig 3. SMOTE representation.**

reduce the error or cost function. Finally, the optimized values are obtained. Lastly, the output layer will generate the "D" and "ND" classifications.

Fig 4 illustrates how the model functions and how data flows. The summary demonstrates how robustness is impacted by the unbalanced data (training data) provided to the SMOTE algorithm. PCA reduces the degree of dimensionality of training data (blue arrows indicate this). The input, reduced in dimension through PCA (indicated by the red arrow), is further processed by an MLP for categorization. Once the model's parameters are fine-tuned, it can make classification predictions. The high-dimensional data was fed into PCA, while the low-dimensional data (PCA output) was used as input for the MLP model to facilitate classification.

**4.2.1 Synthetic Minority Over-sampling Technique (SMOTE).** Previously, SDP models encountered challenges due to highly imbalanced datasets. Small classes were difficult for classification algorithms to detect accurately, necessitating balancing imbalanced datasets for precise SDP. Various balancing techniques exist in the SPAM-XAI model. SMOTE creates artificial data samples and adjusts the distribution of classes by oversampling the tiny class without substituting. This phase's tiny class is oversampled, which involves attributes area activities. K-NN are chosen, and replicas are created based on differences between their NN and feature vectors of the target value. These synthetic samples are then created by inserting the feature vector of interest with a random value between 0 and 1 along a line segment

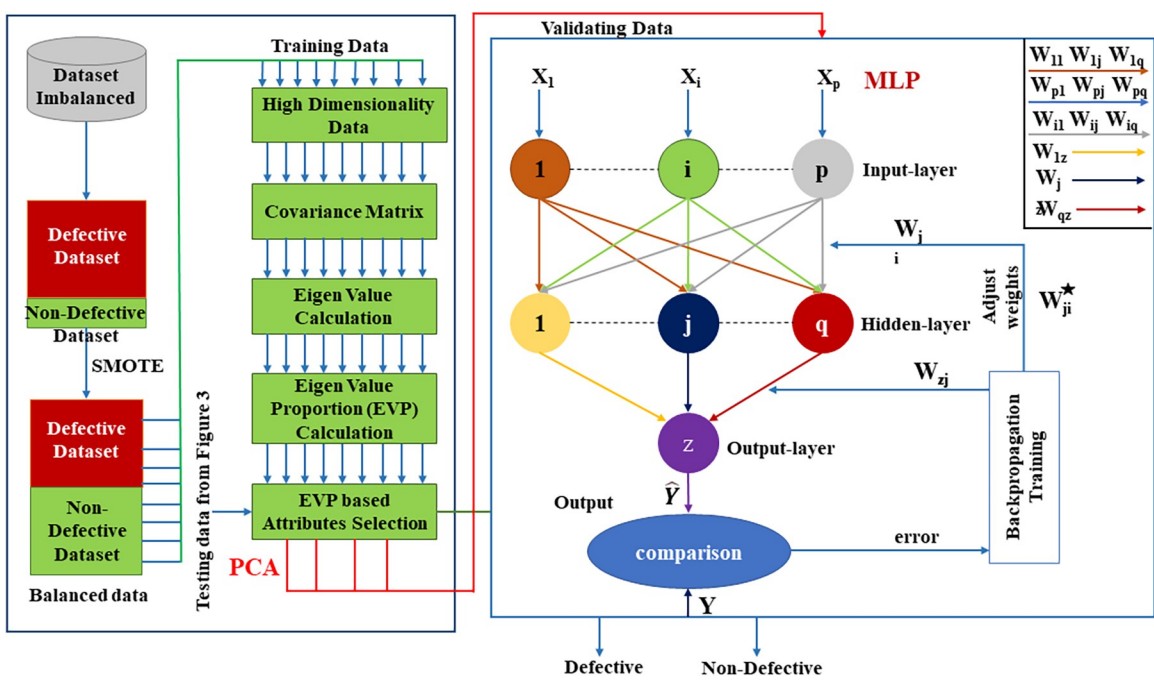

**Fig 4. Internal architecture of the SPAM-XAI model.**

connecting two distinct characteristics.

$$y_{new} = y + rand(0,1) \times (\bar{y} - y) \tag{1}$$

The notation "rand (0, 1)" denotes a random number selected uniformly between 0 and 1."

**4.2.2 Principal Component Analysis (PCA).**   In ML algorithms, the primary task is to find the desired features for the classification problem. Large and small datasets can extract essential features to mitigate overfitting issues by employing feature optimization and dimensionality reduction techniques. In our research, we utilized the PCA technique. Unsupervised ML algorithms, such as PCA and multivariate statistical approaches, aid in reducing the number of variables by selecting the most relevant ones. This technique affects the data by removing fundamental trends from the dataset. It produces a compact set of new multivariate data known as "Principal Components" (PC) by closely tying together strongly related variables. These elements capture a variety of information well. Cross-validation methods like bootstrap and jack-knife are used to assess the efficacy of the PCA model. The following procedures are involved in solving the PC problem mathematically.

- Examine the entire dataset that consists of the $N + 1$ dimension. Use the following formula (Eq 2) to get the average of each dimension for the whole dataset:

$$\bar{x_A} = \frac{1}{n}\sum_{i=1}^{n} x_{Ai} \tag{2}$$

- After that, use the features approach below to generate the covariance matrix for the full dataset. Using Eq 3, $x_A$ and $x_B$:

$$Cov(x_A, x_B) = \frac{\sum_{i=1}^{n} (x_{Ai} - \bar{x}_A)(x_{Bi} - \bar{x}_B)}{n}$$

(3)

- Now, find the eigenvectors and corresponding eigenvalues of the covariance matrix. To do this, solve the characteristic equation (Eq 4):

$$|A - \lambda.I| = 0$$

(4)

Following eigenvector and eigenvalue acquisition, diminish dimensionality and craft a feature vector. The eigenvector linked to the highest eigenvalue signifies the dataset's principal component. Construct a matrix, denoted as M, with dimensions N × I, arranging eigenvectors in a descending order based on eigenvalues. Opt for the I eigenvectors with utmost significance. Project the model into the fresh subspace by applying the N × I eigenvector matrix.

**4.2.3 Multilayer Perceptron (MLP).** There are three types of learning: unsupervised, supervised, and semi-supervised. Classification categorizes them based on addressed problems. In SDP, our focus is classification to predict software defects. ML offers diverse classification methods like DT, LR, and RF. These techniques, foundational in AI, tackle classification challenges. Among supervised ML, ANN stands out. CM1, PC1 and PC2 datasets aid model training and validation. Dataset split precedes metadata initialization, encompassing weights, learning rates, hidden layers, minimum error, and epochs. The sigmoid function, depicted in Fig 5, is commonly employed for classification.

It may be expressed mathematically: Vector defines an input layer that may be a distinct characteristic to identify software defects.

$$X = [x_1, x_2, x_3, \ldots x_i]$$

(5)

Where X represents the input features in the input layer, the output layer, $Z = [Z_1, Z_2]$ will represent the projected class.

$$B = [b_1, b_2, b_3, b_4 \ldots b_i]$$

(6)

Where B represents the weights. The net input ($net_j$) to the j$^{th}$ hidden neuron is

$$net_j = h_{0j} + \sum x_i * b_{ij}$$

(7)

Where $h_{0j}$ is the bias term for the j$^{th}$ hidden neuron. $x_i$ are the input features. $b_{ij}$ are the weights from the i$^{th}$ input features to the j$^{th}$ hidden unit. The $\sum x_i * b_{ij}$ represents the weighted sum of the inputs.

The output $h_j$ of the j$^{th}$ hidden neuron can be represented as:

$$h_j = f(net_j)$$

(8)

$f$ is the activation function applied to the $net_j$.

$$net_k = \sum h_j * v_{jk} + h_{0k}$$

(9)

Where net input ($net_k$) to the k$^{th}$ output neuron. It is the weighted sum of all the outputs from the hidden neurons ($h_j$) multiplied by their respective weights ($v_{jk}$) connecting them to the k$^{th}$

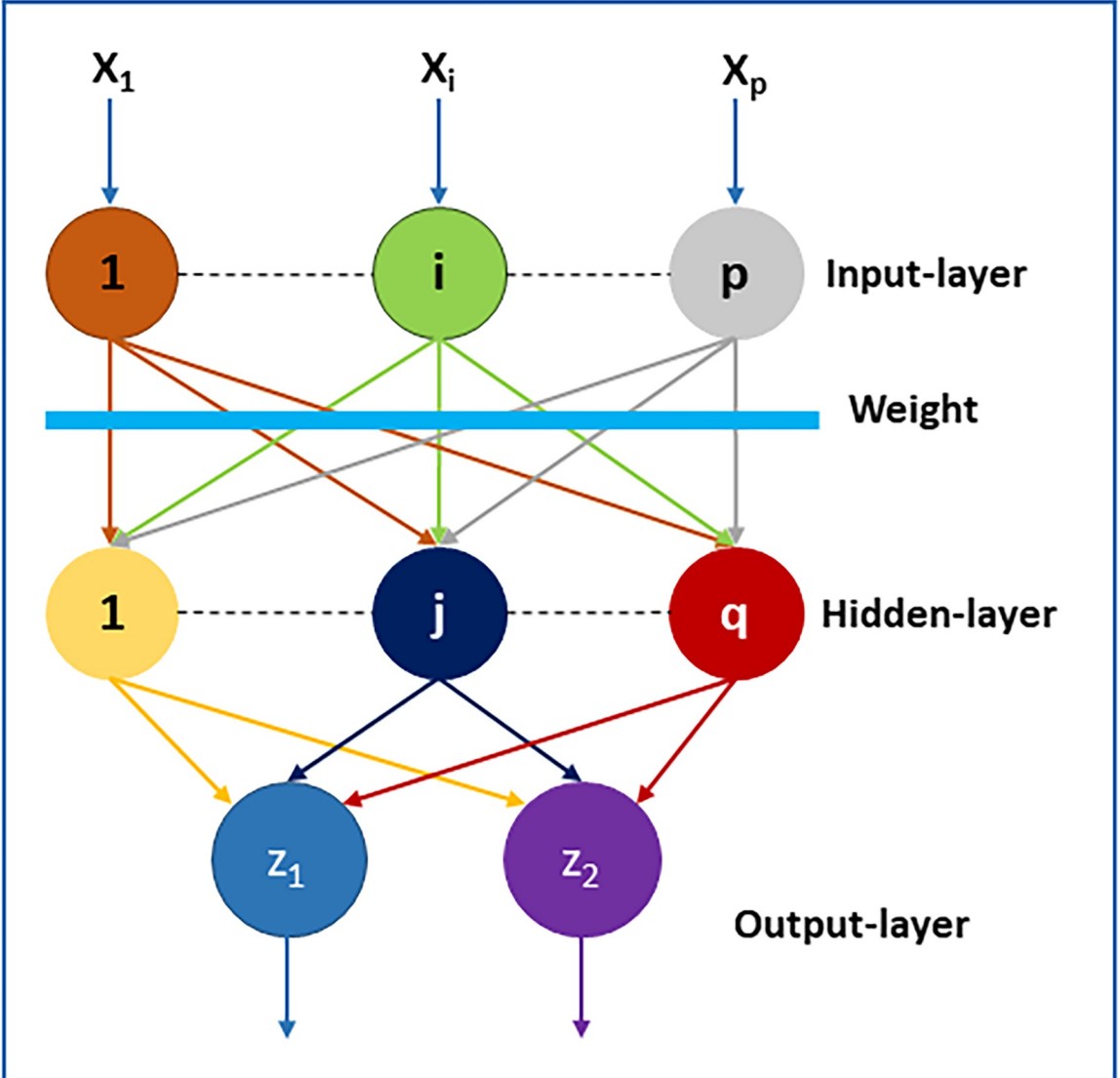

**Fig 5. Illustration of MLP.**

output neuron, plus the bias term ($h_{0k}$) for that output neuron.

$$y_k = g(net_k) \qquad (10)$$

Where output ($y_k$) of the k<sup>th</sup> output neuron, which is obtained by applying an output activation function ($g$) to the net input ($net_k$).

The terms "ND" and "D" can indicate error correction.

$$\delta l = \left( \frac{ND}{D} - y_k \right) * outputlayerderivative\ g(net_k) \qquad (11)$$

The term $\left( \frac{ND}{D} - y_k \right)$ calculates the difference between the target output and the predicted output for a given neuron k *outputlayerderivative g($net_k$)* represents the derivative of the activation function *g* with respect to its input $net_k$.

Updation of the weight can be done as follows:

$$\delta v_{jk} = \beta * \delta l * h_j \qquad (12)$$

where: $\delta v_{jk}$ is the weight change for the connection between the j$^{th}$ hidden neuron and the k$^{th}$ output neuron. β is the learning rate parameter. $\delta l$ is the error signal at the k$^{th}$ output neuron, $h_j$ is the output of the j$^{th}$ hidden neuron.

For k$^{th}$ neuron (Weight Change for Bias).

$$\delta v_{0k} = \beta * \delta l \; for \; bias \qquad (13)$$

Here $\delta v_{0k}$ is the weight change for the bias term associated with the k$^{th}$ output neuron. The bias term is often considered to have a constant input of 1.

We have sent an error backward (Weight update).

$$v_{jk}(new) = v_{jk}(old) + \delta v_{jk} \qquad (14)$$

Where, $v_{jk}(new)$ Updated weight from the j$^{th}$ hidden neuron to the k$^{th}$ output neuron and $v_{jk}(old)$ Previous weight from the j$^{th}$ hidden neuron to the k$^{th}$ output neuron (Bias Weight Update).

$$v_{0k}(new) = v_{0k}(old) + \delta v_{0k} \qquad (15)$$

Here $v_{0k}(new)$ Updated bias weight for the $k^{th}$ output neuron. $v_{0k}(old)$ is the previous bias weight for the $k^{th}$ output neuron.

**4.2.4 LIME (Local Interpretable Model-Agnostic Explanations).** Being an ANN, it provides a relatively recent model that explicitly provides the input's eXplanation. It is a model-agnostic algorithm that can be utilized twofold with any predictor (i.e. classifier or regressor). Such information allowed territories to model the local environmental situation rather than the general plan of the whole empire and region. The simple meaning of LIME is to approximate a compounded model as a simple interpretable model near the input. The algorithm, in turn, peeks at the sample area of the given input and then serves a descriptive model that fits those samples. A complex framework, on the other hand, can be demonstrated by a simple line. Mentioning [55], LIME consists of a description of information networks. In [56], the authors have extended LIME for image classification, which is called LIME-Image get, integrated with LIME for image data, and in [57], the authors have also extended LIME for text classification, which is LIME-Text gets LIME for text data [58]. These researches have demonstrated that LIME can be a legitimate substitute for complex model explainers and help users understand the model structure. A LIME algorithm is based on estimating how the behaviour of the complex model generally performs using a simple linear model. Given a black-box model *f*, LIME compares the model's behaviour by locally approximating it with a linear model *g*, which is described as:

$$g(x) = w0 + w1x1 + w2x2 + \cdots + wn*xn \qquad (16)$$

For x represents an entity, and w is a set of weights. LIME performs the approximation by taking instances from the 'local neighbourhood' around x, called the "local neighbourhood sampling". These instances are then used to train the linear model g, which is learned by solving the following optimization problem:

$$minimize_g ||f(x) - g(x)|| + \Omega(g) \qquad (17)$$

Where f(x) represents the predictions from the complex model, $g(x)$ is a linear function that takes *x* as input and produces an output. Supposing we define $|(|f(x)-g(x)|)|$ as the difference

between the predictions taken from the complex model. The regularization term $\Omega(g)$ typically represents a penalty on the complexity of the linear model $g$, aiming to prevent overfitting and ensure simplicity and interpretability. One common form of regularization is the *L2* regularization, which penalizes the squared values of the coefficients of $g$. Mathematically, it can be expressed as:

$$\Omega(g) = \lambda \sum_{j=1}^{p} \|w_j\|_2^2$$

Where $\lambda$ is the regularization parameter, controlling the strength of regularization. $p$ is the number of features in the linear model $g$. $w_j$ represents the coefficient associated with the $j^{th}$ feature in $g$. $\|.\|_2$ denotes the *L2* norm, which computes the Euclidean norm of a vector. The term $\|w_j\|_2^2$ calculates the squared value of the coefficient $w_j$, and the summation over $j$ ensures that the regularization term penalizes the overall complexity of the linear model.

The LIME algorithm also assigns different weights to the instances based on their proximity to x. These weights are computed using a kernel function, such as the exponential kernel or the radial basis kernel, represented as:

$$K(xi, x) = \exp\left(-\frac{\frac{dxi^2}{dx}}{\sigma^2}\right) \tag{18}$$

In which $x_i$ depicts a sample from the neighbourhood, x is a case to be discussed. $\frac{dxi}{dx}$ indicates a distance between $x_i$ and x, and $\sigma$ stands for a parameter that will determine the width of the kernel. The last stage of LIME consists of assigning meaning to every feature of the linear model g by computing the absolute values of the model weights as the significance measure for each feature. The items fastest moving are.

Below is this flowchart of the SPAM-XAI model in black and white boxes, as shown in Fig 6. The black box uses the ML techniques to predict probable SDP conclusions. Moreover, the XAI is also included, which generates the reasons concerning a specific dataset and attribute influencing the SDP. Besides, the SPAM-XAI model can offer a detailed explanation of the software; thus, the developers, stakeholders, and other project parties can make clear decisions ahead of the development that will reduce the time, resources, and cost of software development.

The black box here refers to our MLP component in Fig 6 above, which uses a neural network architecture for making predictions. The black box pipes in the preprocessed dataset that

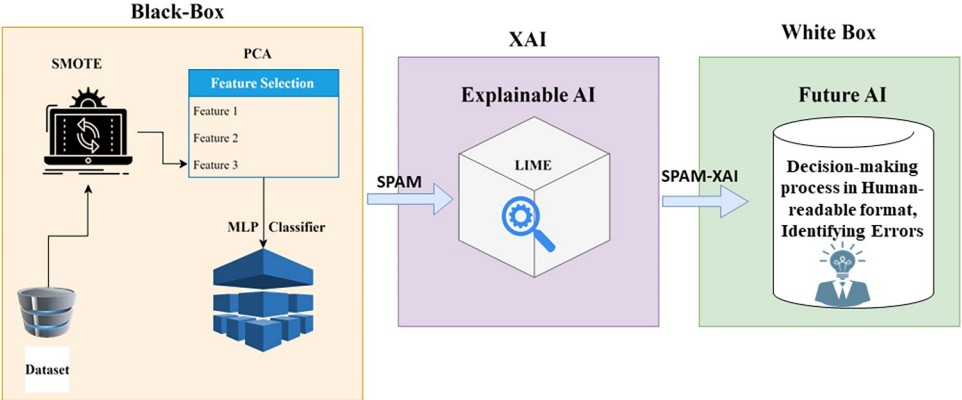

**Fig 6. Overview of SPAM-XAI model complete architecture.**

is obtained through SMOTE and PCA, and the output is that the predicted class label is either "ND" or "D". Within the NN system is the difficult-to-understand portion that is not visible to the user which is why it has been called "black box". The white box displays the LIME part of the model employed in the prediction explanation instead of the black box. LIME produces explanations of high interpretability for complex predictions by identifying and making the parameters that most contributed to the output of the black box explicit. According to that, the white box improves the transparency and interpretability of the black box. It provides users with insights on where the predictions were made and about the potential model errors or biases.

```
# Algorithm (SPAM-XAI)
Start
1. Import necessary libraries          // Import necessary libraries
and files
2. Data = read (defect_dataset)               // Read available
data from a file
3. Data = Train Dataset, Testing Datasets. Divided (Ratio)      //
dividing the data into two groups:
                                a training model and a testing model,
according to the appropriate ratio
4. SMOTE                    // Eq (1), Dataset balance
5. Set the learning rate to arbitrary, initialize the random weights
W, and bias b with any number.
6. Optimized features                  // Decide which characteristics
are relevant for the prediction.
7. Describe and input X              // Eq (5)
8. Compute the deep layer's net input (netⱼ)              // as per
Eq (7)
9. Determine the hidden layer's entire output estimation f(hᵢₙₚᵤₜ)
            //Eq (8)
                                  // Identification of the acti-
vation function is bipolar or binary sigmoidal
10. Calculate the overall input using the result level
hⱼ                            //Eq (8)
11. The total output of the final level g(netₖ)          // Eq
(10)
12. Error calculation δl                  // Eq (11)
13. Error correction & weight Updation                  // Eqs
(11-13)
14. Updates to weights and biases      //Eq (12) Minimal weight
adjustments to determine the
                                      ideal gap
15. Apply LIME algorithm          // Eqs 16-18) to explain the pre-
dictions of the mode
        understand the features that have the most impact on the pre-
dictions made by the model l
16. Fine-tune the model          //by adjusting the parameters of
SMOTE, PCA, MLP, and LIME
17. Repeat Steps 6-16:              // Once epochs are finished,
recalculate by predicting the outcome
                              using the updated W
18. End
19. Evaluate Performance              // To assess the mode's
effectiveness.
20. Accuracy ← (Total no.of correctpredictions)/(Total no.of predictions) *100
21. Output // Providing the model's accuracy as an output
Stop
```

**Table 6. Demonstration confusion matrix.**

|  | Expected Quantities | | |
|---|---|---|---|
|  |  | Positive | Negative |
| **Real Quantities** | **Positive** | True Positive (TP) | False Positive (FP) |
|  | **Negative** | False Negative (FN) | True Negative (TN) |

## 4.3 Metrics evaluation

This section shows the evaluation criteria for the SPAM-XAI model, which involves various methods, including a confusion matrix, classification accuracy, precision, recall, F-m, and AU-Roc curve.

**4.3.1 Purpose of using evaluation metrics.** It is crucial to analyse our model utilizing a variety of measures. Performance measures are essential to ensure that the paradigm functions correctly and adequately.

**4.3.2 Confusion matrix.** The confusion matrix is used to define the performance of the SPAM-XAI model as shown in Table 6.

**4.3.3 Measures of classification.** It is essentially an expanded form of the confusion matrix. Other metrics than the confusion matrix also aid in achieving a more profound comprehension and studying our model's functionality.

- **Accuracy:** There are two types of solutions in a confusion matrix: TP and TN. The authors must practically evaluate the SPAM-XAI model's accuracy for industrial SDP applications. So, being more accurate can help in better decision-making and reduce the cost of testing and efforts.

$$Accuracy = \frac{(TP) + (TN)}{(TP + FN + FP + TN)}$$

- **Precision:** It measures the defects to the total predicted defects in the SPAM-XAI model. Precision in our model must be used to identify both "D" and "ND" items as defective, regardless of whether the classification was accurate or not.

$$Precision = \frac{(TP)}{(TP) + (FP)}$$

- **Sensitivity/Recall:** It measures "D" values, which are all actual defects in the classification. It can detect all the "ND" values in the dataset. Without bothering about how "D" values are incorrectly or correctly differentiated. It can be represented as:

$$Recall = \frac{(TP)}{(TP) + (FN)}$$

- **F1-score:** It employs precision and recall. It is an essential evaluation metric because it sums up the model's predictive performance elegantly. It can vary from 0 to 1. Zero represents the worst possible outcome, and one represents the best. It can be described as:

$$F = \frac{2 * Precision * Recall}{Precision + Recall}$$

- **AU-Roc Curve:** We have used the ROC curve because it gives an appropriate angle when the dataset is balanced. The coordinate of the ROC curve contains two variables; one is the True Positive Rate (TPR), which is the proportion of the truly predicted value that was correctly predicted for all true values. Another variable is the False Positive Rate (FPR), the false predicted value proportion to the total false values.

TPR=(TP/(TP+FN))
FPR=(FP/(TN+FP))

The curve is shown as a function of two variables: Actual Positive Rate (TPR) and Type I Error/Predicted Positive Rate (FPR)

$$FPR = \frac{(FP)}{(FP) + (TN)}$$

## 5. Implementation

This section describes the application of the work by employing SPAM-XAI model for classification. The adoption of the model is done through the application of Python language with Spyder being the Integrated Development Environment (IDE). The classification model SPAM-XAI uses Sklearn – an ML library for Python – as the learning algorithm for this model. During the entire course of the work, the NASA MDP repository is used as a source and venue for various datasets used for training and evaluation. Especially, the model is taught and tested by utilizing these datasets for evaluating the efficiency for SDP. During the implementation phase, some settings are made regarding different parameters aimed at enhancing the efficiency of a model. These parameters are hidden layer sizes '100 50', activation function 'relu', solver 'adam', alpha (regularization parameter) '0. 001", learning rate="0. '001', iter. max='200', momentum="0.9", and epsilon= "1e-08". LimeTabularExplainer: Feature names: [list of feature names], Class names: [list of class names], Kernel width: 0. 25, Feature Selection: Range: "auto", "lasso_path", "forward_selection", and "none", Mode: "classification". These values are chosen through trial and error as well as based on the requirements of the dataset to achieve the best possible performance SPAM-XAI model in SDP tasks.

### 5.1 Dataset CM1

The authors have worked on the CM1 dataset, driven from the NASA dataset repository. The confusion matrix of the SPAM-XAI model is shown in Table 7.

The SPAM-XAI model is done using the CM1 dataset, which is illustrated in Table 7. The testing dataset consisted of 597 sample observations. The model correctly classified "D" instances 571 times (True Positives) and included 10 as "ND" (False Negative). Also, it found that "ND" data was labeled as "D" 16 times more often (False Positive), whereas "ND" was labeled as "D" 0 times (True Negative). It illustrates the high true positive and true negative rates and good performance of the model.

**Table 7. SPAM-XAI confusion matrix.**

|  |  | Expected | |
|---|---|---|---|
|  |  | D | ND |
| **Real** | D | 571 | 10 |
|  | ND | 16 | 0 |

Moreover, the ROC curve contains the model's efficiency evaluation. The depiction is a curve plotted FPR versus TPR, which measures the model's trade-off between sensitivity and specificity. An area under the curve (AUC) value of 0.91 was obtained. Such a high value suggests excellent classification accuracy. Altogether, we conclude that the model displayed the optimal AUC value of 0.91, which successfully confirms the ability to differentiate between positive and negative class labels, as shown in Fig 7.

## 5.2 PC1 dataset

The authors have worked on the PC1 dataset, driven from the NASA dataset repository. The confusion matrix of the SPAM-XAI model is shown in Table 8.

The SPAM-XAI model is done using the PC1 dataset, which is illustrated in Table 8. The testing dataset consisted of 224 sample observations. The model correctly classified "D" instances 215 times (True Positives) and included 2 "D" instances as "ND" (False Negative). Also, it found that "ND" data was labeled as "D" 7 times more often (False Positive), whereas "ND" was labeled as "D" 0 times (True Negative). It illustrates the high true positive and true negative rates and good performance of the model.

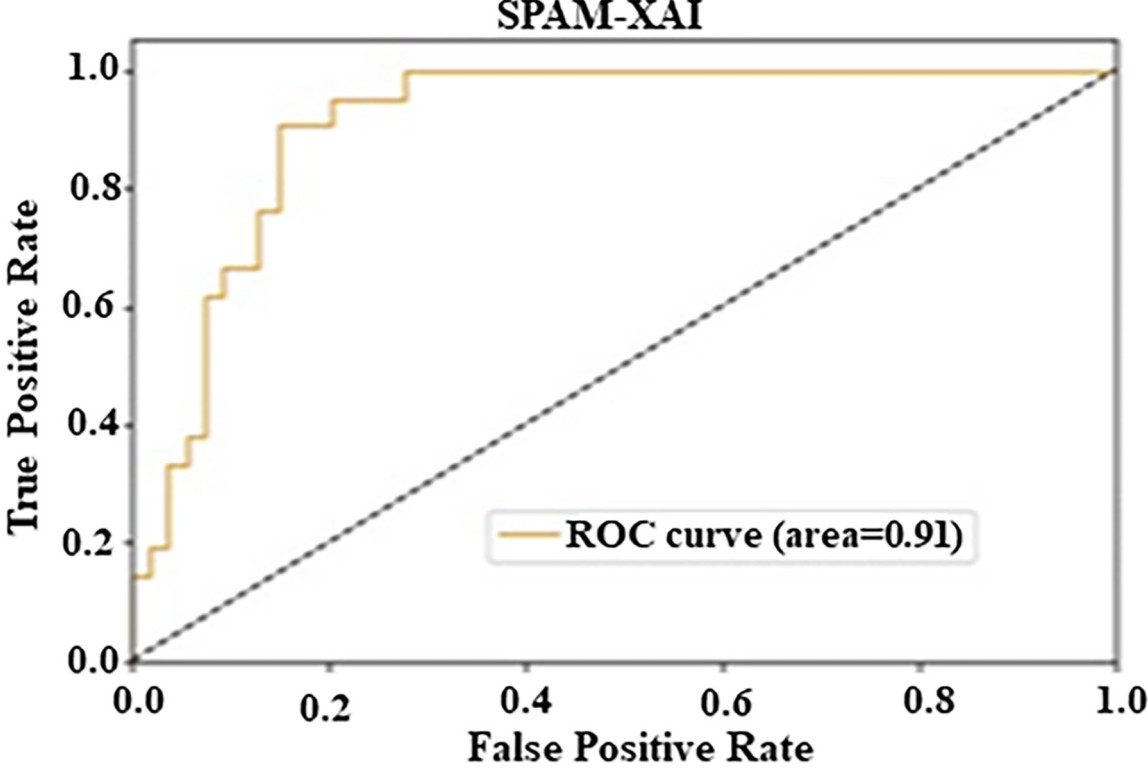

**Fig 7. Analysis of CM1 ROC curve.**

**Table 8. SPAM-XAI confusion matrix using PC1 dataset.**

| | | Expected | |
|---|---|---|---|
| | | D | ND |
| **Real** | D | 215 | 2 |
| | ND | 7 | 0 |

Furthermore, the ROC curve contains the model's efficiency evaluation. The depiction is a curve plotted FPR versus TPR, which measures the model's trade-off between sensitivity and specificity. An AUC value of 0.79 was obtained. Such a high value suggests excellent classification accuracy. Altogether, we conclude that the model displayed the optimal AUC value of 0.79, which successfully testifies to the ability to differentiate between positive and negative class labels, as mentioned in Fig 8.

The authors have evaluated the SPAM-XAI model on various matrices that show the performance of CM1, PC1, and PC2 datasets.

### 5.3 PC2 dataset

The authors researched using the PC2 dataset derived from the NASA dataset repository. The confusion matrix for the SPAM-XAI model is presented in Table 9 of their work.

The test dataset with 212 observations utilized comprising the PC2 dataset is shown in Table 9 below. The SPAM-XAI model was proven right in 186 cases (True Positives). While in

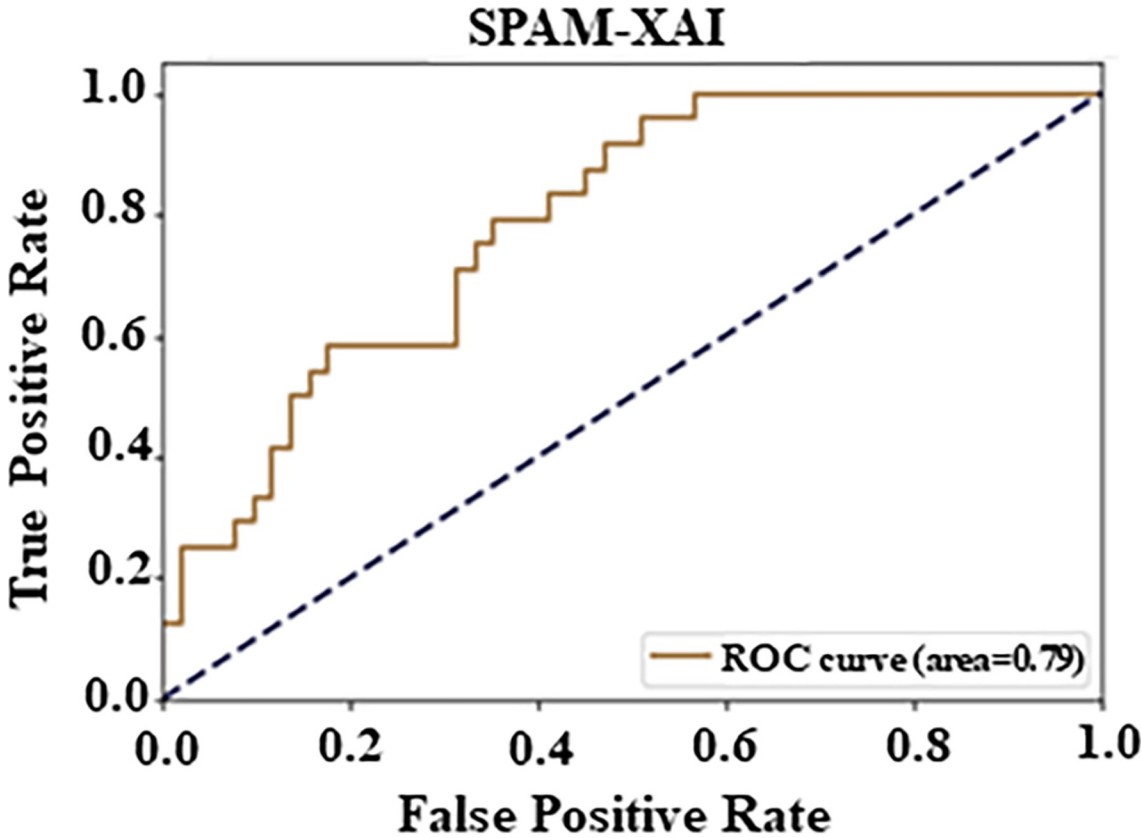

**Fig 8. Analysis PC1 AU-ROC curve.**

**Table 9. SPAM-XAI confusion matrix using PC2 dataset.**

|  |  | Expected |  |
| --- | --- | --- | --- |
|  |  | D | ND |
| **Real** | D | 186 | 13 |
|  | ND | 9 | 4 |

13 cases (False Negatives), it identified the "D" data as "ND". Furthermore, the algorithm identified False Positives ("ND" data as "D") 9 times and True Negatives ("D" declared "ND" data) 4 times. This distribution communicates a strong model performance as it maintains equally good True Positive and True Negative values. Besides that, the ROC curve is an additional evaluation measure of our model, showing our approach's diagnostic ability. The graph concerns the relationship between FPR and TPR and goes along the curve from 0 to 1. Using the 0.5 reference line, we can assess performance. The ROC curve's shape tells the model's sensitivity and specificity levels, and the point with the highest specificity is attentive when the curve approaches the y-axis with 0.2 or lower. The AUC value of 0.59 suggests that the classification ability is satisfactory, greater than 0.5, and lower than 1. To sum up, the results of experiments demonstrated that the model significantly outperforms the PC2; the AUC is close to 0.60, as Fig 9 shows.

Table 10 presents the performance metrics for defect prediction models evaluated on three datasets: CM1, PC1, and PC2, covering Precision, Recall, F-Measure, AU-ROC, cross-validation Accuracy, and validation Accuracy. For CM1, the model achieved a Precision of 95.00%, recall of 96.00%, F-Measure of 95.00%, and AU-ROC of 91.00%, with cross-validation and validation Accuracies of 98.20% and 97.91%, respectively, indicating high consistency and overall performance. The PC1 dataset showed a Precision of 97.01%, recall of 99.00%, F-Measure of

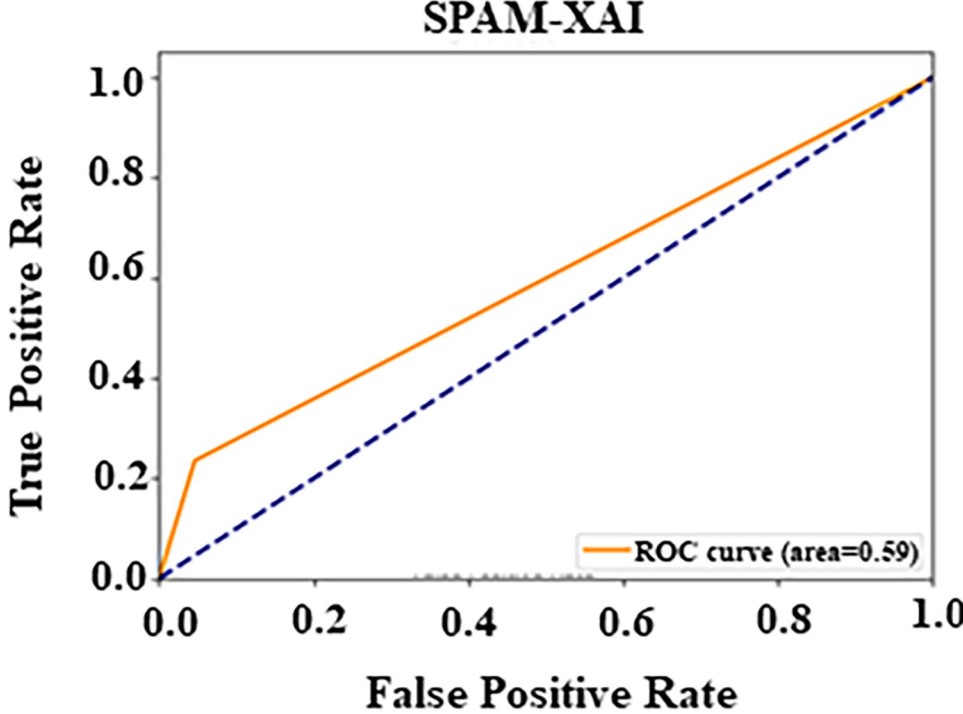

**Fig 9. Analysis PC2 AU-ROC curve.**

**Table 10. Statistical model evaluation for CM1, PC1 & PC2 datasets using SPAM-XAI model.**

| Dataset | Precision | Recall | F-Measure | AU-ROC | Accuracy (cross-valid) | Accuracy (Validation) |
|---------|-----------|--------|-----------|--------|------------------------|------------------------|
| CM1 | 95.00 | 96.00 | 95.00 | 91.00 | 98.20 | 97.91 |
| PC1 | 97.01 | 99.00 | 98.00 | 79.00 | 97.51 | 92.10 |
| PC2 | 94.00 | 97.01 | 95.10 | 59.00 | 99.52 | 98.65 |

98.00%, and a lower AU-ROC of 79.00%, with cross-validation Accuracy at 97.51% and validation Accuracy at 92.10%, still maintaining high performance despite the drop in discrimination ability. For PC2, the model achieved a Precision of 94.00%, recall of 97.01%, F-Measure of 95.10%, and a notably lower AU-ROC of 59.00%. However, it excelled in cross-validation and validation, with an accuracy of 99.52% and 98.65%, respectively, highlighting challenges in distinguishing defect instances while maintaining high accuracy.

# 6. Results and discussion

This section establishes our experimental study on datasets to validate our proposed model on various ML-based measurement scales or evaluation criteria. We have represented our experimental study's analysis, interpretation, and justification by comparing our presented model results with the previously developed approaches.

## 6.1 SPAM-XAI model experimental comparison with baseline models

This section represents the rigorous experimental comparison, and the SPAM-XAI model is pitted against various baseline models, as shown in Table 11.

Fig 10A and 10B shows the graphical representation of the baseline model comparison using CM1 and PC1 datasets for experimental verification of the SPAM-XAI model.

**Table 11. Experimental evaluation of various baseline models with the SPAM-XAI model.**

| Dataset | Baseline Models | Precision | Recall | F-Measure | AU-ROC | Accuracy |
|---------|-----------------|-----------|--------|-----------|--------|----------|
| CM1 | NB | 50.77 | 53.09 | 93.78 | 77.48 | 91.92 |
| | SVM | 97.79 | 50.00 | 92.70 | 69.35 | 97.79 |
| | RF | 55.25 | 56.13 | 93.93 | 87.16 | 97.71 |
| | LR | 50.18 | 51.05 | 91.62 | 49.66 | 97.48 |
| | DT | 62.13 | 66.56 | 90.69 | 66.56 | 96.65 |
| | KNN | 48.89 | 49.96 | 93.66 | 57.68 | 97.71 |
| | LDA | 54.52 | 55.27 | 91.15 | 80.03 | 96.19 |
| | QDA | 58.82 | 64.86 | 94.35 | 72.20 | 93.24 |
| | **SPAM-XAI** | **95.00** | **96.00** | **95.00** | **91.00** | **98.20** |
| PC1 | NB | 58.25 | 60.23 | 87.70 | 74.16 | 87.32 |
| | SVM | 45.69 | 50.00 | 87.28 | 52.80 | 91.39 |
| | RF | 60.14 | 58.27 | 89.37 | 88.07 | 92.05 |
| | LR | 61.35 | 57.92 | 89.07 | 72.08 | 91.39 |
| | DT | 62.04 | 64.20 | 88.23 | 64.20 | 88.39 |
| | KNN | 45.67 | 49.76 | 87.06 | 51.94 | 90.96 |
| | LDA | 62.95 | 59.48 | 88.68 | 83.77 | 90.09 |
| | QDA | 53.31 | 55.96 | 87.65 | 61.04 | 89.90 |
| | **SPAM-XAI** | **97.01** | **99.00** | **98.00** | **79.00** | **97.51** |

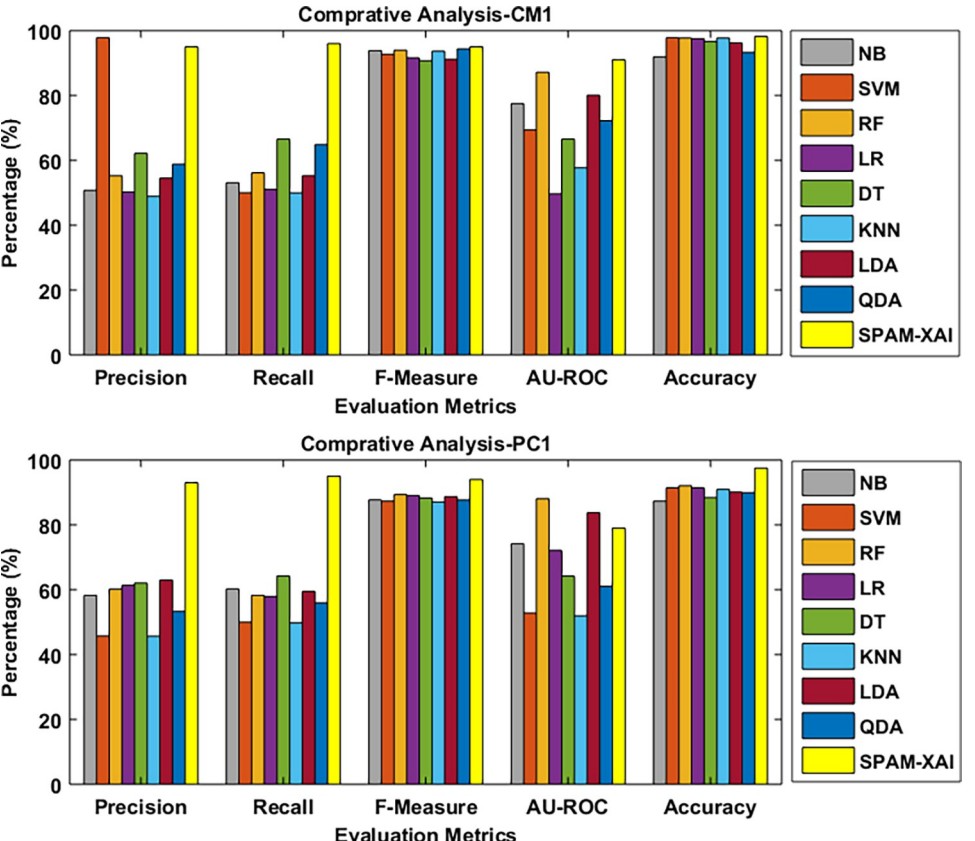

**Fig 10.** **(a).** Baseline model Comparative Analysis with CM1 dataset. **(b).** Baseline model Comparative Analysis with PC1 dataset.

## 6.2 SPAM-XAI model comparison with prior State-of-the-art

The estimation of the performance of the proposed model can be validated by comparing the results, and we have shown the comparative analysis of the proposed model with the previously developed one in Table 12.

Table 12 shows the discussion by comparing various models on CM1 and PC1 datasets. Table 12 shows the discussion by comparing various models on CM1 and PC1 datasets. The SPAM-XAI model is outperforms other models in SDP because of its holistic approach. Thus, the model reduces generalisation and rich prediction errors due to factors such as class imbalance and high-dimensional feature vectors. It also uses advanced XAI methods for model interpretability to avoid ambiguous decisions or lack of trust in the model's predictions. Carefully fine-tuned hyperparameters further enhance its effectiveness and decrease its computational cost. However, there may be some previously developed models that may possess some evaluation metrics that outperform the SPAM-XAI model due to some domain-specific features, appropriate algorithm selection, data, and evaluation biases. These models may perform well in certain aspects depending on their comparative strengths in terms of alignment with certain characteristics of a given dataset or on the selection of some evaluation metrics as opposed to others and showcasing the importance of considering multiple factors in model evaluation and selection.

**Table 12. Analysis of previously developed models with SPAM-XAI model.**

| Dataset | Models | Precision | Recall | F-m | Accuracy | AU-ROC |
|---------|--------|-----------|--------|-----|----------|--------|
| CM1 | Naïve Bayes [49,36] | 86.20 | 78.65 | 34.09 | 64.57 | 70.0 |
| | Random Forest [49,58] | 71.10 | 71.29 | 32.17 | 60.98 | 76.0 |
| | C4.5 Miner [37,59] | 74.91 | 74.66 | 27.68 | 66.71 | 53.0 |
| | Immunos [37,60] | 73.65 | 75.02 | 30.99 | 66.03 | 63.0 |
| | ANN-ABC [37,60] | 75.00 | 81.00 | 33.00 | 68.00 | 77.0 |
| | HSOM [38,60] | 70.12 | 78.96 | 30.65 | 72.37 | 80.0 |
| | SVM [39,49] | 81.20 | 79.08 | 31.27 | 78.69 | 50.0 |
| | Majority vote [40,60] | 79.80 | 80.00 | 30.46 | 77.01 | 81.0 |
| | AntMiner+ [40,60] | 80.65 | 78.88 | 30.90 | 73.43 | 84.0 |
| | ADBBO-RBFNN [59,41] | 81.92 | 80.96 | 29.71 | 82.57 | 90.0 |
| | NN GAPO + B [42] | - | - | - | 74.40 | - |
| | DT [39,50] | 83.30 | 74.23 | 81.20 | 73.49 | 37.0 |
| | KNN [59] MLP [59] | 83.90 90.40 | 84.70 95.50 | 84.30 92.90 | – 86.73 | 47.0 63.4 |
| | HybridModel (PC-SVM) [2] | 96.10 | 99.00 | 97.00 | 95.20 | - |
| | Hybrid approach(GA-DNN) [2] | 80.32 | 97.32 | 89.09 | 97.50 | - |
| | GWOFS-MLP Model [2] | 98.27 | 86.36 | 91.93 | 92.72 | 81.60 |
| | **SPAM-XAI model** | **95.00** | **96.00** | **95.00** | **98.20** | **91.0** |
| | Naïve Bayes [59] | 96.00 | 90.00 | 97.20 | 90.30 | 75.0 |
| | Random Forest [59] | 97.00 | - | 98.80 | 97.69 | 73.0 |
| | C4.5 Miner [60] | 76.58 | 81.76 | 34.05 | 62.18 | 68.0 |
| | Immunos [60] | 81.99 | 79.66 | 36.92 | 61.73 | 64.0 |
| PC1 | ANN-ABC [60] | 89.00 | 83.00 | 33.00 | 65.00 | - |
| | HSOM [60] | 86.79 | 85.67 | 35.67 | 95.87 | - |
| | SVM [60] | 80.98 | 86.59 | 98.00 | 92.45 | 50.0 |
| | Majority vote [60] | 84.61 | 84.37 | 30.98 | 92.50 | - |
| | AntMiner+ [60] | 89.34 | 87.12 | 26.11 | 91.85 | - |
| | ADBBO-RBFNN [60] | 90.89 | 89.33 | 20.24 | - | - |
| | DT [59] | 97.00 | - | 98.00 | - | 57.0 |
| | KNN [59] MLP [59] | 95.00 97.00 | 90.00 99.00 | 98.00 98.00 | 95.71 96.00 | 49.0 74.0 |
| | DNN [2] | 94.00 | 99.00 | 97.00 | 93.00 | - |
| | GWOFS-MLP Model [2] | 99.24 | 95.65 | 97.40 | 95.03 | 84.00 |
| | **SPAM-XAI model** | **97.01** | **99.00** | **98.00** | **97.51** | **79.0** |

## 6.3 A combined comparison analysis using the CM1 and PC1 datasets with the previously created model

This section represents the graphical implementation of the combined comparative study of previously developed models. The illustrations of Fig 11 are shown below.

The graph shown in Fig 11A illustrates that our model precision score (**95.00% with CM1 and 97.01% with PC1**) performs better than the others. The Recall value graph can be represented using Fig 11B. It includes the proposed model recall value (red and blue horizontal line) (**CM1: 96.00 and PC1: 99.00**) percentage, which is higher than the other models.

The graph in Fig 11C illustrates that the SPAM-XAI model F-measure score (**95.00%with CM1 and 98.00%with PC1**) performs better than the others. The accuracy value graph can be represented using Fig 11D, which shows that the proposed model's accuracy value (**CM1: 97.00% and PC1: 96.00%**) percentage is higher than that of the other models.

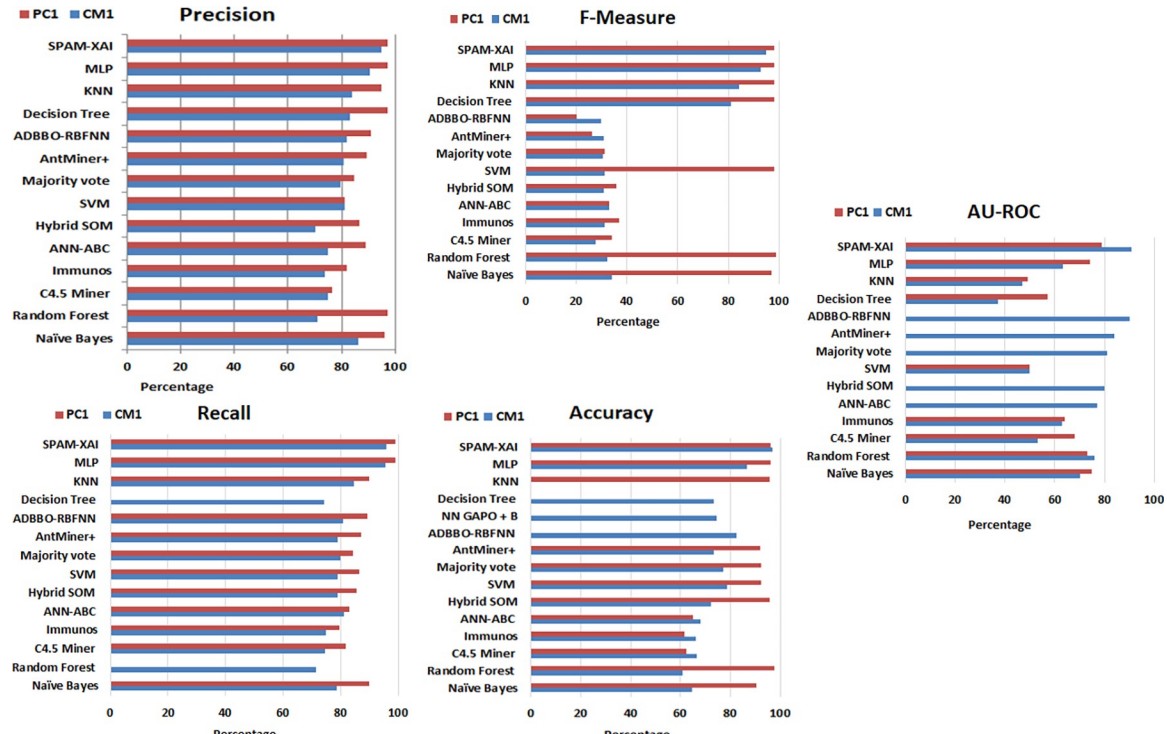

**Fig 11. (a).** Precision CM1 and PC1 dataset. **(b).** Recall CM1 and PC1 dataset. **(c).** F-Measure CM1 and PC1 dataset. **(d).** Accuracy CM1and PC1 dataset. **(e).** AU-ROC-area CM1 and PC1 dataset.

Furthermore, the AU-ROC curve is a significant criterion for validating a model. The AU-ROC curve, as shown in Fig 11E (**91.0% with CM1 and 79.00% with PC1**), is higher than the previously developed one.

## 6.4 Time complexity and computational cost

In this section, we will discuss the time features of the SPAM-XAI model. We will focus on the execution time, which encompasses three main stages: the processing time for the SMOTE phase for data balancing, the MLP training stage, the LIME algorithm implementation, and any other possible stage of processing. This may include variety of operations such as manipulation of the data, transformations or computations. Furthermore, we will discuss the testing time, which directly relates to the time used for inferences or predictions. With this timing analysis, we will seek to bring insights into how the SPAM-XAI model is efficient in time, revealing useful information about how time is budgeted for each crucial stage of its use.

*Total execution time*
$$= EL\ Phase(Time) + SDA\ Phase(Time) + Additional\ Processing(Time) + Testing(Time)$$

$$Total\ Training\ Time = Total\ SPAMXAI\ Execution(Time) - Testing(Time)$$

Where the SDA phase time involves Stacked Denoising Autoencoder (SDA), Ensemble Learning (EL) is the training stage.

SPAM-XAI model is created with the intention of being useful for making predictions while also being as computationally efficient as possible. We show the runtime complexity of the model, training time, comparison of time cost with other state-of-the-art models [2,61].

**Table 13. Comparison of total training duration (in seconds) of previously developed models with SPAM-XAI.**

| Dataset | SPAM-XAI | GWOFS-MLP | BPDET | RF | MLP | L-SVM-B | Bagging | NB | AdaBoost |
|---------|----------|-----------|-------|-----|-----|---------|---------|-----|----------|
| CM1 | 4.2589 | 5.2132 | 12.64 | 0.091 | 1.331 | 0.021 | 0.031 | 0.0001 | 0.051 |
| PC1 | 3.1257 | 2.5321 | 16.53 | 0.212 | 2.370 | 0.031 | 0.061 | 0.0001 | 0.061 |

The SPAM-XAI model further highlight the focus on performance and computational efficiency. This comprises of the discussion regarding the temporal aspects of the model such as the time complexity of the model, the total duration required to train the model and lastly a comparison between the temporal costs in terms of other models are discussed in Table 13.

Table 13 below shows the performance of the SPAM-XAI model where it can be noted that during training SPAM-XAI consistently outperforms other state-of-the-art models in terms of training time. Overall the SPAM-XAI model is most efficient in computational performance compared to others. This lower time cost can be ascribed to the optimal feature selection using the MLP and the data balancing using SMOTE which were both model-efficient. This explains the significance of SPAM-XAI in time costs and it is therefore recommended as a novel approach for situations where computational effectiveness is high on priorities.

## 7. eXplainability- AI (XAI)

XAI is used in this model to give interpretability and transparency to the SPAM-XAI model predictions. A LIME algorithm is used to explain the model's predictions. The LIME algorithm generates local interpretations of a prediction by locally fitting an interpretable linear model on the input space. The algorithm operates by giving noise to the input data and monitoring the changes in the model's predictions.

### 7.1 CM1 dataset

The SPAM-XAI model interprets the ML model's outcomes as an application and a technique. Among those things is identifying the critical features that affect the probability of issues and handling how attribute values impact the prognosis. The SPAM-XAI model aims to provide a platform for software developers and engineers to give them guidelines on dealing with the source of the mess. This is achieved by providing transparency and unambiguous insight into the dataset. Fig 12 depicts the importance of sub-class attributes in SDP. For instance, the prediction production of SPAM-XAI on CM1 dataset output is shown in Fig 12. The result indicates prediction probabilities for the given sample: 0% for Irreparable and 100% for No Repairs at All. The model guarantees that the instance is defect-free with certainty at this high condensability. The subsequent section briefly details the feature set-up and corresponding values, which is critical in making the correct prediction. Among all the features, CALL_-PAIRS has a power of -1.16, which has the highest correlation with the prediction, and the power weight is only 0.33. One of the most distinguishing features is LOC_BLANK, which has a value of -3.94 and is the importance of 0.7, indicating its effect. Varying other metrics, for example, the NO_OF_BRANCHES, DECISION_DEPTH, DECISION_COUNT, LOC_CODE_BRANCH, COUINIT_METRIC, CYCLOMATIC_DENSITY, INCIDENT_COUNT, and FACADE_LATERAL_SIZE play their role but to a much lesser extent the product provides the features' values and their weights that were used to generate the prediction ultimately. In this case, the lack of any defect occurs because the model has analysed all the attributes involved and seen how each attribute contributes to the result.

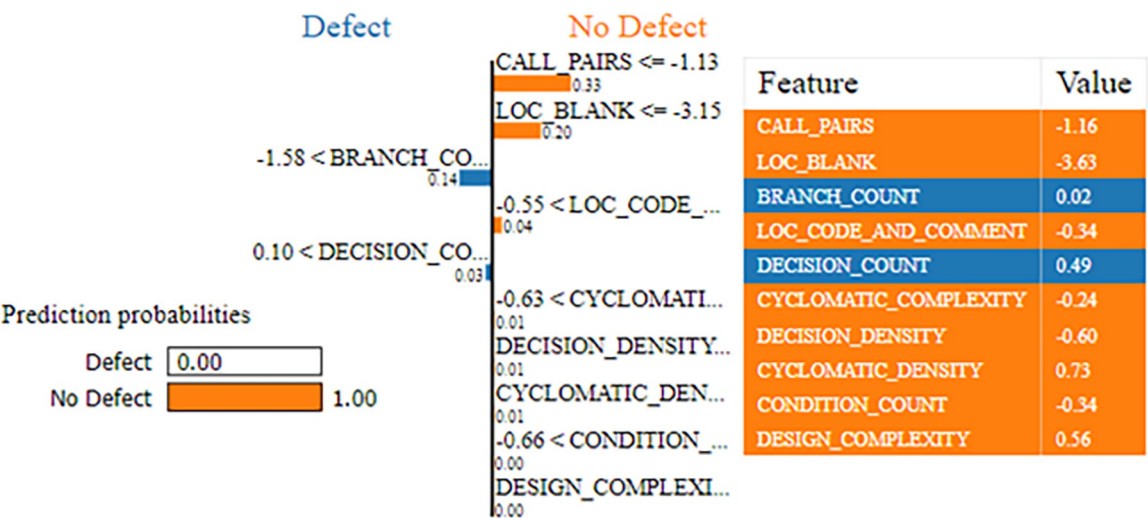

**Fig 12. SPAM-XAI using the CM1 dataset.**

## 7.2 PC1 dataset

The SPAM-XAI model is shown to be valuable in the exploration of the ML model trained with the PC1 dataset. SPAM-XAI model predictions via locally fitting a linear multivariate regression counterpart. In addition, the SPAM-XAI model could enhance the examination of the relevance of particular features with the help of the feature importance scores. The model acquires better interpretability through this enhancement and cultivates the user's confidence and faith, as shown in Fig 13.

The predicted model of the sample (Fig 13) shows that the sample was an "ND" module with a probability of occurrence of 0.99, and a "D" module had an occurrence probability of only 0.01. The main component of the metric is "HALSTEAD_CONTENT" with 15.83, along with "CYCLOMATIC_DENSITY" and "PARAMETER_COUNT" having 0.20 and 0.00, respectively. These are the impacts of each feature on the prediction's result, which can be seen by using the SPAM-XAI model as the explainability model of the algorithm. We would imitate the local model behaviour guiding us around the predicted instance and will provide a humanly interpretable explanation for the model predictions.

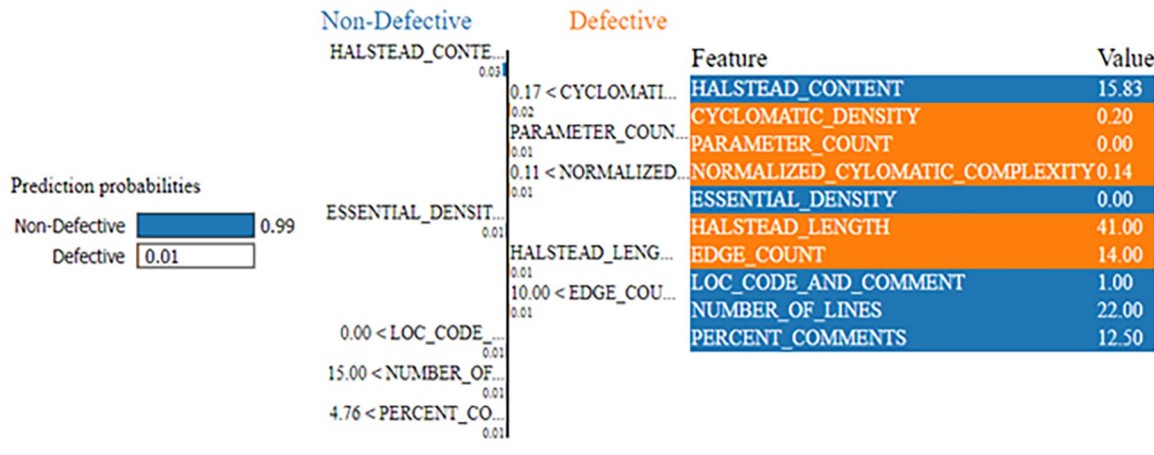

**Fig 13. SPAM-XAI using the PC1 dataset.**

## 8. Conclusion and future direction

The SPAM-XAI model is a state-of-the-art hybrid tool that combines the best features of ML and deep learning algorithms. This novel approach effectively addresses the weaknesses that have limited the efficiency of previous state-of-the-art models. Some of the most important limitations are the ones that correlate imbalanced datasets, the curse of dimensionality, the influence of outliers, and the explainability of the models. SPAM-XAI, which utilises CM1, PC1, and PC2 datasets, has shown its transformative capability during the experimentation phase, with the results being concluded. On the CM1 dataset, its performance yields striking metrics: accuracy (97.00%), recall (96.00%), precision (95.00%), AU-Roc (91.0%), and F-measure (95.00%). Besides this, the SPAM-XAI model is efficient on the PC1 dataset with accuracy (96.0%), recall (99.00%), precision (97.01%), F-measure (98.00%), and AU-Roc of 79.0%. In addition, on the PC2 dataset, where it accomplished great accuracy (98.65%), precision (94.00%), recall (97.01%), F-measure (95.10%), and AU-Roc (59.00%), respectively. This hybrid model significantly improves precision by 10.2% on the CM1 dataset and enhances source code quality, reliability, and maintainability. SPAM-XAI offers an optimized, cost-effective solution for software development, making the SDP process more precise and efficient and bringing a new perspective to defect prediction in the earlier stages of development. The proposed model excels in computational efficiency.

Future Direction: Considering what is still to be learned and what to investigate more in the SPAM-XAI model, research and exploration are exciting.

- Firstly, the model's capabilities by submitting it to a variety of larger-in-scope datasets already as diverse as software contexts can show a lot about the depth of its adaptability and robustness in software environments. Providing this means of assessment will be advantageous for the researcher to ensure his investigation of the limitations and superiority of the model in real-life scenarios.

- Secondly, algorithmic diversity would be another direction to grow the model's catalogue. An ML environment with diverse algorithms and XAI models (for comparison) would be enlightening to see their strong and weak sides. This could be beneficial for appreciating the SPAM-XAI model characteristics in the overall framework. Such comprehensive assessment would practically answer the medical staff's demand for what model to apply.

- The final extension of the model will be to different phases of the SDLC, allowing us to explore fascinating insights. Considering its operation in various stages in the project's scope as to be specific to requirements engineering, design, and maintenance would allow for the emergence of innovative applications and opportunities for its improvement.

This thorough evaluation may give the other team members an understanding of how the model could influence software quality and reliability during development.

## Author Contributions

**Conceptualization:** Mohd Mustaqeem, Mahfooz Alam.

**Data curation:** Mohd Mustaqeem.

**Formal analysis:** Mohd Mustaqeem, Suhel Mustajab, Mahfooz Alam, Fathe Jeribi, Shadab Alam, Mohammed Shuaib.

**Funding acquisition:** Fathe Jeribi.

**Methodology:** Suhel Mustajab, Mahfooz Alam, Mohammed Shuaib.

**Project administration:** Fathe Jeribi, Shadab Alam.

**Resources:** Suhel Mustajab, Fathe Jeribi, Shadab Alam.

**Supervision:** Suhel Mustajab, Fathe Jeribi, Shadab Alam.

**Validation:** Mohammed Shuaib.

**Visualization:** Mohammed Shuaib.

**Writing – original draft:** Mohd Mustaqeem, Mahfooz Alam.

**Writing – review & editing:** Suhel Mustajab, Fathe Jeribi, Shadab Alam, Mohammed Shuaib.

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
