## [Decision Letter · Decision Letter 0]

17 May 2024

PONE-D-24-11437A Trustworthy Hybrid Model for Transparent Software Defect Prediction: SPAM-XAIPLOS ONE

Dear Dr. Jeribi,

Thank you for submitting your manuscript to PLOS ONE. After careful consideration, we feel that it has merit but does not fully meet PLOS ONE’s publication criteria as it currently stands. Therefore, we invite you to submit a revised version of the manuscript that addresses the points raised during the review process.

We look forward to receiving your revised manuscript.

Kind regards,

Hikmat Ullah Khan, PhD (Computer Science)

Academic Editor

PLOS ONE

 [The authors extend their appreciation to the Deputyship for Research& Innovation, Ministry of Education in Saudi Arabia, for funding this research work through the project number ISP-2024.].  

[The authors extend their appreciation to the Deputyship for Research& Innovation, Ministry of Education in Saudi Arabia, for funding this research work through the project number ISP-2024.]

 [The authors extend their appreciation to the Deputyship for Research& Innovation, Ministry of Education in Saudi Arabia, for funding this research work through the project number ISP-2024.]

6. Please remove your figures from within your manuscript file, leaving only the individual TIFF/EPS image files, uploaded separately. These will be automatically included in the reviewers’ PDF.

Additional Editor Comments :

Following concerns needs the authors attention:

1) The paper promises to be about XAI for SPAM detection. However, there are not interpretation of reasons behind prediction of results? So what is eXplainable here. All methods are like KNN NB etc which are common ML algorithms.

2) Data Imbalancing is the issue which is shared as research gap. however, nature of SPAM is about imbalance class. So, it is an imbalanced data problem inherently. obviously if i receive 1000 emails on gmail, only few will be SPAM so it has to be imblanced like MEDICAL diagnositc etc. so why to balance it?

3) PseudoCode is very poorly presented, better remove it , framework is good enough.

4) XAI is added in framework but it does not show how it AI has become XAI?

5) no need of sharing what is confusion matrix

6) equation numbers are missing and many other basic format issues.

Reviewers' comments:

Reviewer's Responses to Questions

**Comments to the Author**

1. Is the manuscript technically sound, and do the data support the conclusions?

Reviewer #1: Yes

Reviewer #2: No

2. Has the statistical analysis been performed appropriately and rigorously? 

Reviewer #1: N/A

Reviewer #2: No

3. Have the authors made all data underlying the findings in their manuscript fully available?

Reviewer #1: Yes

Reviewer #2: Yes

4. Is the manuscript presented in an intelligible fashion and written in standard English?

Reviewer #1: Yes

Reviewer #2: No

5. Review Comments to the Author

Reviewer #1: 1. Revise Reference Numbering

2. In the Introduction part, strong points of this proposed method should be further stated and organization of this whole paper is supposed to be provided in the end.

3.The Introduction is not enough.

4.The articles listed in References are old. (Most of them are out of date. For example, the papers published within 2 years are used to calculate CiteScore in SCOPUS.) The authors should survey past studies in detail. Besides, the authors should justify the effectiveness of the proposed method by comparing with state-of-the-art methods.

5.The authors should compare the proposed method with other researchers’ methods. There are many previous works. The authors should emphasize the difference with other researchers’ methods. Add more comparison data.

6.The problem definition of this work is not clear. In Sect. 2, the drawbacks of each technique should be described one by one. The authors should emphasize the difference with other methods to clarify the position of this work further.

7. Figures are not clear

8.Revise all abbreviations

9.Analysis of complexity of the algorithm both theoretically and empirically must be added (Space and Time Complexity)

Reviewer #2: 1. The paper lacks an introduction about software defects. It is necessary to include a section that defines software defects and discusses their nature and types to provide context and enhance the reader's understanding of the issues addressed.

2.Figures 2 and 3 are incomplete. Please ensure that all figures are fully developed and include all necessary labels, legends, and descriptions to effectively convey the intended information.

3.Equations 7, 8, 12 and 17 are incomplete. Please provide the full equations with all necessary terms and explanations to ensure clarity and accuracy.

4.Certain terms in Equation 17 are not defined in the paper. Please include the missing equations and provide clear definitions and explanations for each term to ensure the equation is fully understandable.

5.The paper's title mentions that the algorithm is transparent; however, this justification is not clearly provided in the text. Please elaborate on how and why the algorithm is considered transparent, including details on its design, implementation, and the ways it allows for easy understanding and verification by others.

6.For Tables 1, 2, and 3, please ensure that parameterized comparisons are included to facilitate a more thorough analysis. Additionally, any multi-line text should be moved to the supporting sections to maintain the clarity and conciseness of the tables

7.The validation of the algorithm should be conducted in real-time scenarios rather than solely relying on a portion of the dataset. Real-time validation will provide a more accurate assessment of the algorithm's performance in practical use cases and enhance its credibility and applicability.

8.Please specify which stage of the SDLC the algorithm is incorporated into. Elaborate on how integrating the algorithm at this stage impacts the overall SDLC process, considering a real-time scenario. Providing insights into how the algorithm affects requirements gathering, design, implementation, testing, or maintenance phases will enhance the understanding of its integration within the SDLC framework.

6. PLOS authors have the option to publish the peer review history of their article (what does this mean?). If published, this will include your full peer review and any attached files.

Reviewer #1: No

Reviewer #2: No

---

## [Author Response · Author response to Decision Letter 0]

23 May 2024

We have incorporated the necessary changes and suggestions from the editor and reviewers into the revised version and attached a separate response to review comments and details of corrections made. Please see the attached file for more details.

---

## [Decision Letter · Decision Letter 1]

9 Jun 2024

PONE-D-24-11437R1A Trustworthy Hybrid Model for Transparent Software Defect Prediction: SPAM-XAIPLOS ONE

Dear Dr. Jeribi,

Thank you for submitting your manuscript to PLOS ONE. After careful consideration, we feel that it has merit but does not fully meet PLOS ONE’s publication criteria as it currently stands. Therefore, we invite you to submit a revised version of the manuscript that addresses the points raised during the review process. Please submit your revised manuscript by Jul 24 2024 11:59PM. If you will need more time than this to complete your revisions, please reply to this message or contact the journal office at plosone@plos.org. Please include the following items when submitting your revised manuscript:A rebuttal letter that responds to each point raised by the academic editor and reviewer(s). You should upload this letter as a separate file labeled 'Response to Reviewers'.A marked-up copy of your manuscript that highlights changes made to the original version. You should upload this as a separate file labeled 'Revised Manuscript with Track Changes'.An unmarked version of your revised paper without tracked changes. You should upload this as a separate file labeled 'Manuscript'.If applicable, we recommend that you deposit your laboratory protocols in protocols.io to enhance the reproducibility of your results. Protocols.io assigns your protocol its own identifier (DOI) so that it can be cited independently in the future. For instructions see: https://journals.plos.org/plosone/s/submission-guidelines#loc-laboratory-protocols. Additionally, PLOS ONE offers an option for publishing peer-reviewed Lab Protocol articles, which describe protocols hosted on protocols.io. Read more information on sharing protocols at https://plos.org/protocols?utm_medium=editorial-email&utm_source=authorletters&utm_campaign=protocols.

We look forward to receiving your revised manuscript.

Kind regards,

Hikmat Ullah Khan, PhD (Computer Science)

Academic Editor

PLOS ONE

Journal Requirements:

Additional Editor Comments:

The issues raised are properly addressed/clarified.

Reviewers' comments:

Reviewer's Responses to Questions

**Comments to the Author**

1. If the authors have adequately addressed your comments raised in a previous round of review and you feel that this manuscript is now acceptable for publication, you may indicate that here to bypass the “Comments to the Author” section, enter your conflict of interest statement in the “Confidential to Editor” section, and submit your "Accept" recommendation.

Reviewer #1: All comments have been addressed

Reviewer #2: All comments have been addressed

2. Is the manuscript technically sound, and do the data support the conclusions?

Reviewer #1: Partly

Reviewer #2: Yes

3. Has the statistical analysis been performed appropriately and rigorously? 

Reviewer #1: I Don't Know

Reviewer #2: Yes

4. Have the authors made all data underlying the findings in their manuscript fully available?

Reviewer #1: Yes

Reviewer #2: Yes

5. Is the manuscript presented in an intelligible fashion and written in standard English?

Reviewer #1: Yes

Reviewer #2: Yes

6. Review Comments to the Author

Reviewer #1: (No Response)

Reviewer #2: both table 7 and table 8 presents the confusion matrix for the proposed model's performance evaluation. The True Negative (TN) value represents the number of instances where the model correctly identifies non-defect software codes as non-defect. In this case, the TN value is reported as 0 indicating that there were zero instances of non-defect software codes being correctly identified by the model.I appreciate your diligence in verifying the accuracy of the explanations.

7. PLOS authors have the option to publish the peer review history of their article (what does this mean?). If published, this will include your full peer review and any attached files.

Reviewer #1: No

Reviewer #2: No

---

## [Author Response · Author response to Decision Letter 1]

11 Jun 2024

The manuscript has been corrected as per the editor's comments, and a separate response to review comments has been attached.

---

## [Decision Letter · Decision Letter 2]

1 Jul 2024

A Trustworthy Hybrid Model for Transparent Software Defect Prediction: SPAM-XAI

PONE-D-24-11437R2

Dear Dr. Jeribi,

We’re pleased to inform you that your manuscript has been judged scientifically suitable for publication and will be formally accepted for publication once it meets all outstanding technical requirements.

Kind regards,

Hikmat Ullah Khan, PhD (Computer Science)

Academic Editor

PLOS ONE

Additional Editor Comments (optional):

Reviewers' comments:

Reviewer's Responses to Questions

**Comments to the Author**

1. If the authors have adequately addressed your comments raised in a previous round of review and you feel that this manuscript is now acceptable for publication, you may indicate that here to bypass the “Comments to the Author” section, enter your conflict of interest statement in the “Confidential to Editor” section, and submit your "Accept" recommendation.

Reviewer #1: All comments have been addressed

Reviewer #2: All comments have been addressed

2. Is the manuscript technically sound, and do the data support the conclusions?

Reviewer #1: Partly

Reviewer #2: Yes

3. Has the statistical analysis been performed appropriately and rigorously? 

Reviewer #1: N/A

Reviewer #2: (No Response)

4. Have the authors made all data underlying the findings in their manuscript fully available?

Reviewer #1: No

Reviewer #2: Yes

5. Is the manuscript presented in an intelligible fashion and written in standard English?

Reviewer #1: Yes

Reviewer #2: Yes

6. Review Comments to the Author

Reviewer #1: (No Response)

Reviewer #2: (No Response)

7. PLOS authors have the option to publish the peer review history of their article (what does this mean?). If published, this will include your full peer review and any attached files.

Reviewer #1: No

Reviewer #2: No
